# Integrated neural dynamics of sensorimotor decisions and actions

**David Thura**[¤a], **Jean-François Cabana**[¤b], **Albert Feghaly**[¤c], **Paul Cisek***

Groupe de recherche sur la signalisation neurale et la circuiterie, Department of Neuroscience, Université de Montréal, Montréal, Québec, Canada

¤a Current address: Lyon Neuroscience Research Center (CRNL) - ImpAct team, Inserm U1028 – CNRS UMR5292 – University of Lyon 1, 69675 Bron, France
¤b Current address: Integrated Regional Center of Cancerology (CRIC), Lévis, Québec, Canada
¤c Current address: Institute for Research in Immunology and Cancer of the University of Montréal, Montréal, Québec, Canada
* paul.cisek@umontreal.ca

**Data Availability Statement:** The data and MATLAB code necessary to reproduce the analyses

## Abstract

Recent theoretical models suggest that deciding about actions and executing them are not implemented by completely distinct neural mechanisms but are instead two modes of an integrated dynamical system. Here, we investigate this proposal by examining how neural activity unfolds during a dynamic decision-making task within the high-dimensional space defined by the activity of cells in monkey dorsal premotor (PMd), primary motor (M1), and dorsolateral prefrontal cortex (dlPFC) as well as the external and internal segments of the globus pallidus (GPe, GPi). Dimensionality reduction shows that the four strongest components of neural activity are functionally interpretable, reflecting a state transition between deliberation and commitment, the transformation of sensory evidence into a choice, and the baseline and slope of the rising urgency to decide. Analysis of the contribution of each population to these components shows meaningful differences between regions but no distinct clusters within each region, consistent with an integrated dynamical system. During deliberation, cortical activity unfolds on a two-dimensional "decision manifold" defined by sensory evidence and urgency and falls off this manifold at the moment of commitment into a choice-dependent trajectory leading to movement initiation. The structure of the manifold varies between regions: In PMd, it is curved; in M1, it is nearly perfectly flat; and in dlPFC, it is almost entirely confined to the sensory evidence dimension. In contrast, pallidal activity during deliberation is primarily defined by urgency. We suggest that these findings reveal the distinct functional contributions of different brain regions to an integrated dynamical system governing action selection and execution.

## Introduction

During natural behavior, we are continuously interacting with a complex and dynamic world [1,2]. That world often does not wait for us to make up our minds about perceptual judgments or optimal choices, and inaction can lead to lost opportunities, or worse. Furthermore, we must often make decisions while we are already engaged in an action, such as while navigating

shown in this paper are available at https://doi.org/10.6084/m9.figshare.20805586.

**Funding:** This work was supported by Canadian Institutes of Health Research (https://cihr-irsc.gc.ca/e/193.html) grants MOP-102662 and PJT-166014, the Canadian Foundation for Innovation (https://www.innovation.ca/), Fonds de Recherche en Santé du Québec (https://frq.gouv.qc.ca/en/), the EJLB Foundation (www.ejlb.qc.ca) to PC, and fellowships from the FYSSEN Foundation (http://www.fondationfyssen.fr/en/) and the Groupe de Recherche sur le Système Nerveux Central (https://www.grsnc.org/home) to DT. The funders had no role in study design, data collection and analysis, decision to publish, or preparation of the manuscript.

**Competing interests:** The authors have declared that no competing interests exist.

**Abbreviations:** BIC, Bayesian information criterion; dlPFC, dorsolateral prefrontal cortex; DR, delayed reach; DT, decision time; GPFA, gaussian process factor analysis; GMM, gaussian mixture model; GPe, globus pallidus externus; GPi, globus pallidus internus; M1, primary motor cortex; OT, opposite target; PCA, principal components analysis; PMd, dorsal premotor cortex; PT, preferred target; RT, reaction time; SAT, speed–accuracy trade-off; SP, success probability; TME, Tensor Maximum Entropy; UGM, urgency-gating model.

through our environment or playing a sport [3,4]. These considerations suggest that the neural mechanisms involved in selecting and executing actions should be closely integrated within a unified sensorimotor control system [5]. Indeed, many neural studies have shown considerable overlap between the brain regions involved in action selection and sensorimotor control [6–10].

However, while the anatomical overlap between the distributed circuits of decision-making and sensorimotor control is well established, theoretical models of these processes remain largely separate. Decision-making is often modeled as the accumulation of evidence until a threshold is reached [11–19], at which time a target is chosen. Models of movement control usually begin with that chosen target, toward which the system is guided through feedback and feedforward mechanisms [20–22]. But if the neural circuits involved in action selection and sensorimotor control are truly as unified as neural data suggest, then theories of these processes should be similarly unified. One promising avenue toward an integrated account of selection and control is to consider both as aspects of a single distributed dynamical system, which transitions from a biased competition between actions [23–28] into an "attractor" that specifies the initial conditions for implementing the chosen action through feedback control [29–31].

Here, we test whether neural activity in key cortical and subcortical regions exhibits properties that would be expected from a unified dynamical system for action selection and sensorimotor control. We focus on cells recorded in monkey dorsal premotor (PMd) and primary motor cortex (M1), which are implicated in both selection and control [8,10,32–36], as well as the dorsolateral prefrontal cortex (dlPFC), which is implicated in representing chosen actions [37]. In addition, we examine activity in the output nuclei of the basal ganglia, the globus pallidus externus (GPe) and internus (GPi), whose role in selection and/or motor control is under vigorous debate [38–41]. Importantly, we examine the activity of all of these regions recorded in the same animals performing the same reach selection task, making it possible to quantitatively compare activities in different brain areas using the same metrics.

To disentangle neural activity related to deliberation, commitment, and movement, we trained monkeys to perform the "tokens task" (Fig 1) (see Methods). In the task, the subject must guess which of two targets will receive the majority of tokens jumping randomly from a central circle every 200 ms (Fig 1A). The subject does not have to wait until all tokens have jumped but can take an early guess, and after a target is reached, the remaining tokens jump more quickly (every 150 ms or every 50 ms in separate "Slow" and "Fast" blocks of trials). Thus, subjects are faced with a speed–accuracy trade-off (SAT)—to either wait to be confident about making the correct choice or to take an early guess and save some time, potentially increasing their overall reward rate. If we assume that commitment occurs shortly before movement onset, then we can delimit within each trial a period of deliberation (Fig 1B) during which neural activity should correlate with the sensory evidence related to token jumps as well as to subjective policies related to the SAT. Furthermore, because we can precisely quantify the success probability (SP) associated with each choice after every token jump, we can compute for each trial a temporal profile of the sensory evidence and categorize trials into similarity classes (Fig 1C), including "easy trials," "ambiguous trials," and "misleading trials" (see Methods for details).

Previous studies have shown that both human and monkey behavior in the tokens task is well explained by the "urgency-gating model" (UGM) [14,16,42], which suggests that during deliberation, the sensory evidence about each choice (provided by the token distribution) is continuously updated and combined with a nonspecific urgency signal, which grows over time in a block-dependent manner, and commitment to a given choice is made when the product of these reaches a threshold.

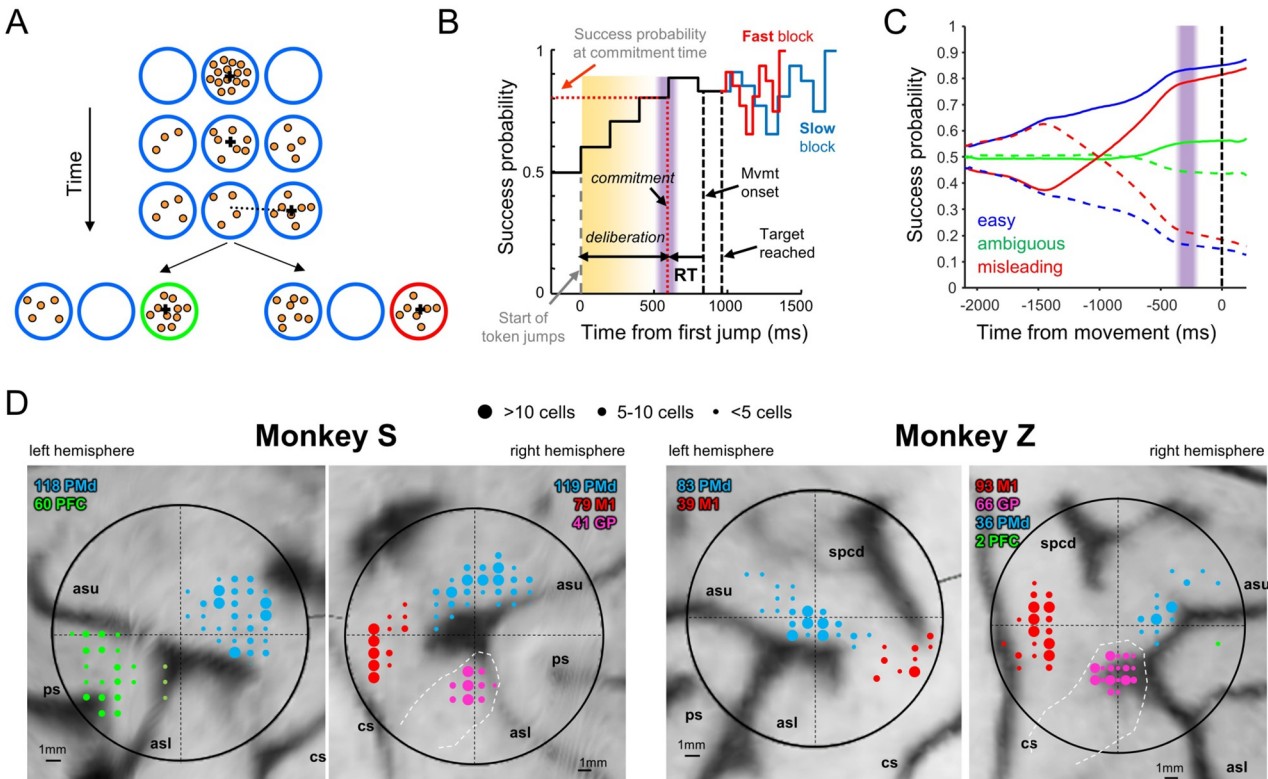

**Fig 1.** (**A**) During each trial of the "tokens task," 15 tokens jump, one every 200 ms, from the central circle to one of two outer target circles. The subject's task is to move the cursor (black cross) to the target that will ultimately receive the majority of the tokens. (**B**) Temporal profile of the "success probability" that a given target is correct. Once a target is reached, the remaining token jumps accelerate to one every 150 ms ("Slow" block) or 50 ms ("Fast" block). We subtract from movement onset the mean reaction time (RT), measured in a separate delayed-response task, to estimate commitment time (purple bar) and the success probability at commitment time (dotted red horizontal line). (**C**) Success probability for choosing the target on the right, in trial types defined on the basis of the success probability profile (see Methods), here computed after aligning to movement onset (vertical black dashed line). The vertical purple bar indicates estimated commitment time. Solid curves: correct target on the right; dashed curves: correct target on the left. (**D**) Recording locations. Medial is up. Dashed white lines indicate the estimated location of the globus pallidus. asl, lower limb of the arcuate sulcus; asu, upper limb of the arcuate sulcus; cs, central sulcus; ps, principal sulcus; spcd, superior precentral dimple.

Recent neural recordings [36,43,44] have largely supported these proposals, but many questions remain. Are "decision-related" neurons part of a module for choosing a target, which sends its output to a separate module of "movement-related" neurons? Is commitment determined by the crossing of a neural "threshold"? Do the basal ganglia contribute to the deliberation process [39,45–49], or do they simply reflect a choice taken in cortical regions [50,51] and contribute only to movement execution [52]? Answering these questions is difficult given the heterogeneity of cell properties [8,32,53–56] and their apparently continuous distribution along rostro-caudal gradients [54,57] or cortical layers [58]. This leads one to consider whether, instead of serial modules, action selection and execution are two modes of a unified recurrent system distributed across the frontoparietal cerebral cortex and associated basal ganglia/thalamic loops.

Here, we test this proposal by examining activity across all of the regions we recorded in the tokens task (PMd, M1, GPe, GPi, and dlPFC), but without a priori classifying cells into putative functional categories. Instead, we use the neural space approach pioneered in recent years [59–74], in which the entire system is described as a point in a very high-dimensional space defined by the activity of all recorded cells, and then reduced into a lower-dimensional

representation that reveals the main factors governing cell activity across the system. We then perform specific analyses to characterize how activity unfolds during deliberation and commitment, comparing the dynamics of different regions, and quantify to what extent cell properties cluster into distinct functionally interpretable roles. Previous studies have used similar techniques to examine whether specific neural populations exhibit distinct [75] versus mixed selectivity [76–78] to various task-related variables. Here, we ask about the relationships between decision-making and movement. In particular, models with serial decision and action stages predict that neural activity will be separable into components related to deliberation and other components related to movement initiation/execution. In contrast, recurrent attractor models predict a single population in which all of these factors are mixed in a continuum. Some of these results have previously appeared in abstract form [79–82].

## Results

### Neural data

We recorded spiking activity from a total of 736 well-isolated individual neurons in the cerebral cortex and basal ganglia of two monkeys (S and Z), recorded at the locations shown in Fig 1D. Of these, 356 were recorded in PMd (237 from monkey S), 211 in M1 (79 from monkey S), 62 in dlPFC (60 from monkey S), 51 in GPe (19 from monkey S), and 56 in GPi (22 from monkey S). The properties of some of these neurons have been reported in previous publications, focusing on tuned activity in PMd and M1 and the basal ganglia [36,43,44,83]. Here, we additionally include neurons recorded in dlPFC, which, for technical reasons, were almost exclusively obtained in monkey S and to date only described in abstract form [79,80].

### Dimensionality reduction of neural activity into principal components

To examine how the activity of this entire population evolves over time during the task, we performed principal components analysis (PCA) on cells recorded in all five brain regions, including any well-isolated neuron that was recorded in both Slow and Fast blocks (see Fig A in S1 Text for a schematic describing our approach). This included a total of 637 neurons, including 291 in PMd, 195 in M1, 53 in dlPFC, 45 in GPe, and 53 in GPi. In principle, there are many ways one can perform PCA. For example, one approach is to perform PCA separately on neurons from each region, analyzing the dynamics of each independently. Alternatively, one can perform PCA on all cells together, producing a "loading matrix" of coefficients (from each neuron to each PC) that allows one to "project" each population into the space of the same PCs. There are also many ways to align the data (on the start of the trial versus movement onset), different time periods one can include (only before movement, only after, or both), and different ways of grouping trials (into large averaged groups versus specific subtypes). Importantly, as shown in Fig B-E in S1 Text), our results were remarkably robust regardless of these choices and held up when applied to data from each monkey separately, consistently leading to the same conclusions. Using factor analysis and gaussian process factor analysis (GPFA) [67] produced very similar results, so for simplicity, here we focus on PCA because it is the simplest and most readily interpretable.

Here, we show analyses performed using PCA on data aligned on movement onset (because we are interested in the dynamics of commitment) performed on all cells together (because we seek quantitative comparisons of the same PCs across regions). Note that because we recorded many more neurons in PMd and M1 than in the other regions, the variance explained was dominated by the PMd/M1 dynamics. However, reducing the number of cells to an equal number in each region did not change the results apart from making them noisier and changing the percentage of variance explained by individual PCs (Fig Ca in S1 Text). Furthermore,

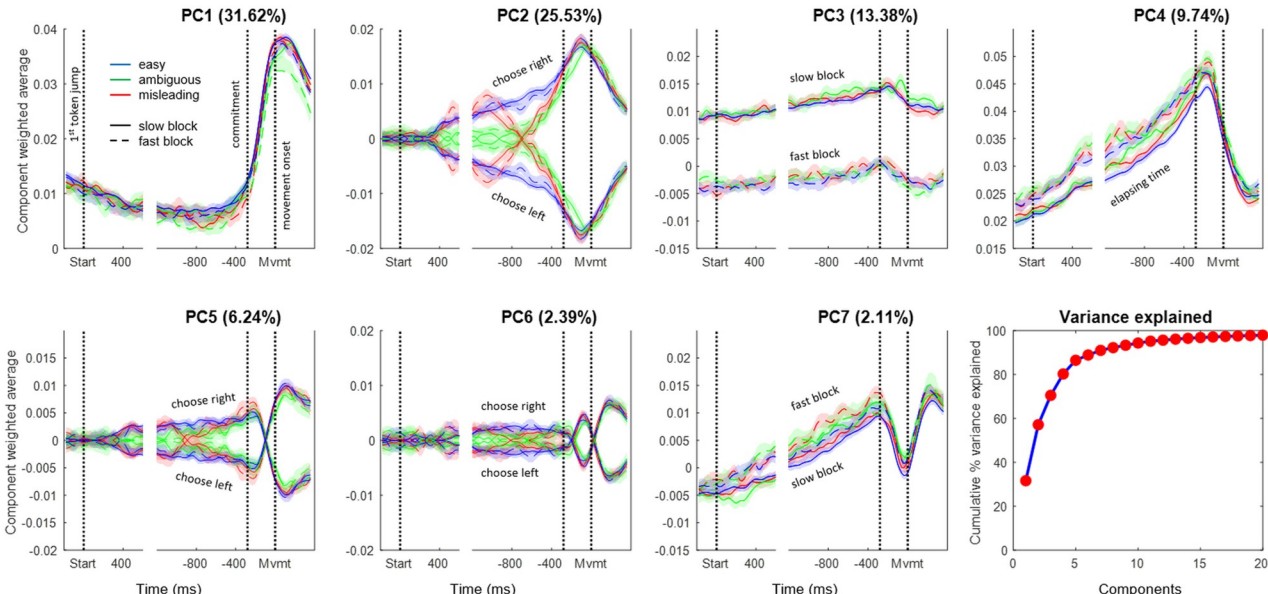

**Fig 2. Components produced by PCA.** The cumulative variance explained by the first 20 components is shown at the bottom right, and the temporal profiles of the top 7 PCs are shown in the rest of the figure. Each of those 7 panels shows the average activity of all cells, weighted by their loading coefficient onto the given PC, for 12 trial groups, including combinations of 3 trial types: easy (blue), ambiguous (green), and misleading (red); two blocks: Slow (solid) and Fast (dashed); for choices made to the left or right (indicated for the components where they differ). Note that the sign is arbitrary because the loading matrix can have positive or negative values. Shaded regions indicate 95% confidence intervals computed using bootstrapping. In each panel, data are shown aligned to the start of token jumps as well as to movement onset. Commitment time is estimated to be 280 ms before movement onset, based on our prior studies [36, 44]. Each panel is scaled to have the same range in the y-axis. PCs are built from the 402 cells that have trials in all conditions. Data and code available at https://doi.org/10.6084/m9.figshare.20805586.

we provided the PCA algorithm with trials grouped into only four large groups—right and left choices in the fast and slow blocks—and let further analyses reveal whether more specific subtypes (easy, ambiguous, etc.) can nevertheless be extracted from the results. Finally, we imposed symmetry on our population by following the "anti-neuron" assumption, which assumes that for every neuron we recorded, there exists a similar neuron with the opposite relationship to target direction (even for cells that are not tuned), effectively doubling the number of neurons. See the Methods section for the justification and motivation for this approach, and Fig E in S1 Text for analyses showing that this procedure does not alter our conclusions.

The first 20 PCs together explained 97.9% of the variance in activity over time across the four groups of trials (Slow-Left, Slow-Right, Fast-Left, and Fast-Right) used for the PCA. Fig 2 shows the temporal profile for the first 7 PCs, separately for easy, ambiguous, and misleading trials in both slow and fast blocks, for both left and right choices, each constructed as a weighted average of the 402 cells that contained trials in all of these conditions (Group 3; see Methods).

The first 4 components, which together explain 80.3% of total variance, are clearly interpretable in terms of the key elements of the UGM. The first PC (31.6% of variance) is nearly identical across all conditions and reflects the transition between deliberation (prior to commitment) and action (after movement onset). It is similar to the main condition-independent component reported in other neural space studies in both primates [84–88] and rodents [76,89,90]. The second PC (25.5%) exhibits two phases. Prior to commitment, it reflects the time course of the sensory evidence on which the monkey made his choice, distinguishing easy, ambiguous, and misleading trials. After commitment, it simply reflects the choice made,

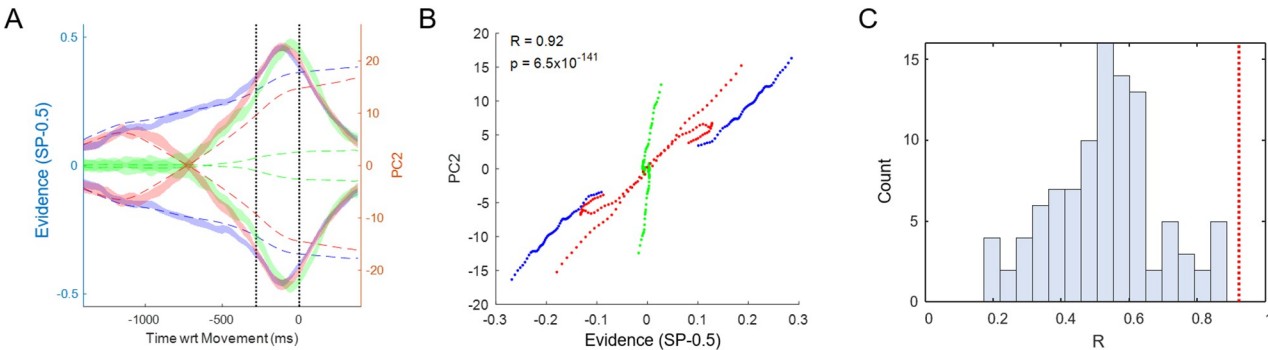

**Fig 3. Comparison between PC2 and sensory evidence.** (**A**) Dashed lines show the sensory evidence in easy (blue), ambiguous (green), and misleading trials (red) calculated as the difference between success probability for the right target and 0.5. Shaded ribbons show the mean and 95% confidence interval of PC2 in the same trials (Slow block, Group 1 cells). The first vertical dotted line indicates commitment and the second indicates movement onset, on which all data are aligned. The evidence trace is delayed by 300 ms, which provides the best fit. Note that until the moment of commitment, the pattern of PC2 closely resembles the evidence, except for diverging toward one of the choices even in the absence of evidence during ambiguous trials. (**B**) The same data, *prior to commitment*, plotted as evidence versus PC2 in these six trial conditions. The correlation coefficient is R = 0.9234 and *p*-value is well below 0.001. (**C**) The distribution (*N* = 100) of correlation coefficients obtained by performing the same analysis on surrogate data sets generated using the Tensor Maximum Entropy approach (see Methods). Here, each surrogate data set is represented by the highest correlation coefficient of any of the top 10 PCs against the profile of evidence. The mean R is 0.5294 (s.t.d. = 0.1615). For comparison, the red line shows the R value from the real data (panel B), and it is higher than all of the R values from surrogate data (*p* < 0.01). Data and code available at https://doi.org/10.6084/m9.figshare.20805586.

without distinguishing trial types. The third PC (13.4%) reflects the block-dependent aspect of urgency, distinguishing between the slow and fast blocks (even before the start of the trial). The fourth PC (9.7%) reflects the time-dependent aspect of urgency until just before movement onset. The remaining components are similar to PCs 2 and 4 during deliberation but capture some of the heterogeneity across cell activity patterns after commitment and movement onset. We discuss these higher PCs in the S1 Text.

PC2 warrants special attention. Note that until the moment of commitment, it correlates very well with the evidence provided by the token movements. Indeed, as shown in Fig 3A and 3B, the correlation between the time-delayed evidence and the value of PC2 is highly significant ($p < 10^{-100}$) with a correlation coefficient of R = 0.92. This is notable because the PCA algorithm was not given any information about these different trial types (easy, ambiguous, misleading, etc.) but was simply given data averaged across four large trial groups that only distinguished left versus right movements and slow versus fast blocks. Nevertheless, when the resulting temporal profiles of PC2 are calculated for specific trial types they clearly reflect how the evidence dynamically changes over the course of deliberation in those trials.

Could this finding be a trivial consequence of overall cell tuning? To test this possibility, we used the Tensor Maximum Entropy (TME) method of Elsayed and Cunningham [91] to generate synthetic data sets that retain primary features such as tuning, but are otherwise random (see Methods), and processed them in the same way as the real data, including cell duplication. As shown in Fig 3C, when PCA is applied to such synthetic data sets, it does not produce components that are as well correlated with evidence as PC2 from our true data ($p < 0.01$). The implication is that the emergence of PC2 in the real neural data requires a consistent relationship between how cells reflect the final choice (left versus right) and how they reflect the evidence that leads to that choice during deliberation (easy versus ambiguous versus misleading, etc.).

Fig 4A and 4B shows the trajectories of the different trial types, separately for the Slow and Fast blocks, plotted in the space of PCs 1, 2, and 4 (PC3 was not used because it is

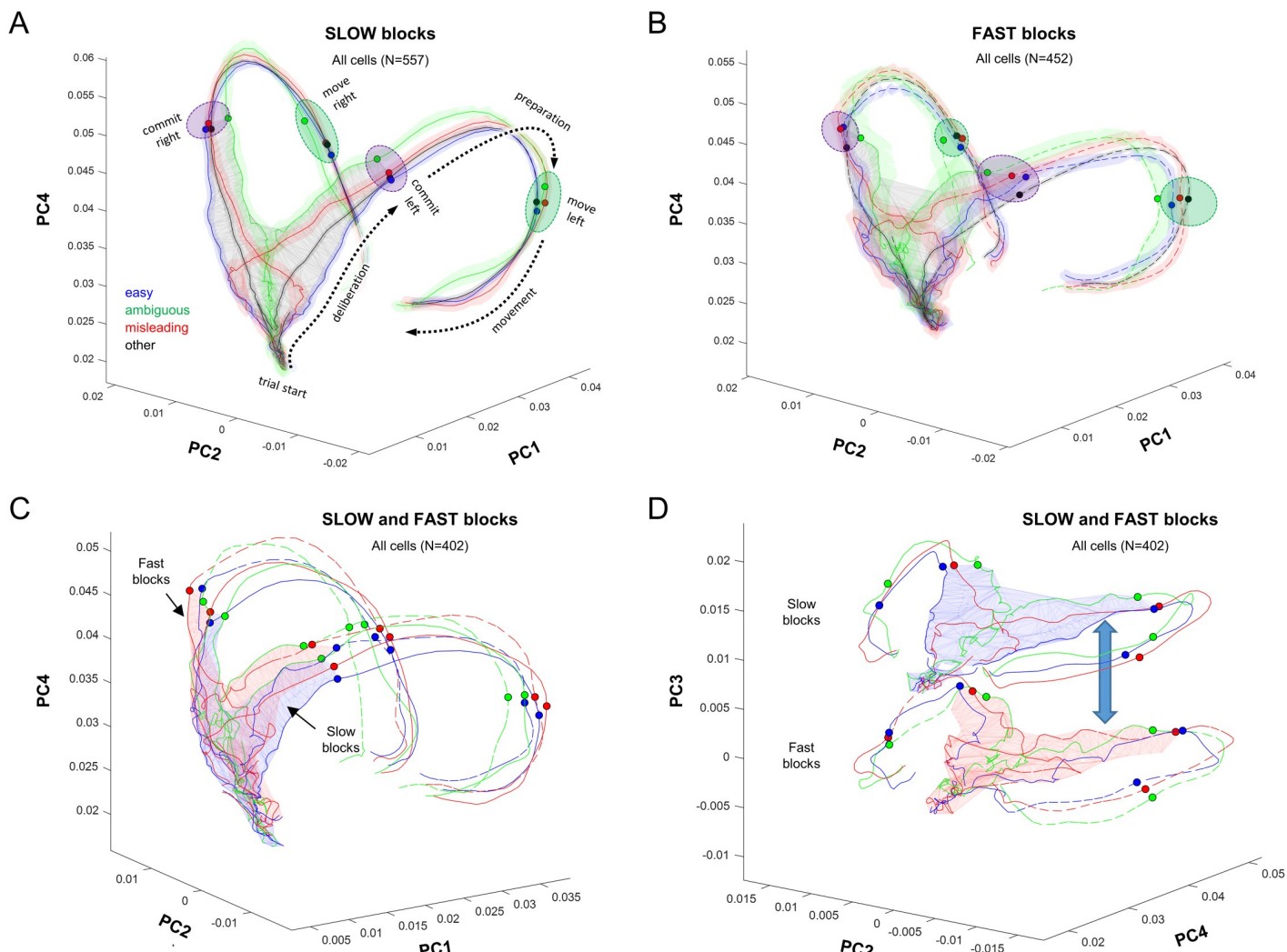

**Fig 4. Neural trajectories in the space of PCs 1, 2, and 4, constructed separately for easy, ambiguous, and misleading trials as well as the other (unclassified) trials.** Each panel shows data aligned on movement and computed from 1,400 ms before to 400 ms after, as well as data aligned on the first token jump and computed from 200 ms before to 500 ms after (using the same loading matrix). (**A**) Data from slow blocks (Group 1 cells). Shaded colored regions around each trajectory indicate the 95% confidence interval computed using bootstrapping. Dotted arrows indicate how the state of activity evolves over time. The gray wireframe encloses all states visited during the deliberation epoch (the "decision manifold"), across all trial types ($\Psi = 0.235$). Purple ellipses indicate the region in which commitment occurs (indicated for individual trial types by small colored circles), and green ellipses indicate the point at which movement is initiated. (**B**) Data from fast blocks (Group 2 cells). $\Psi = 0.403$. (**C**) Comparison of the neural trajectories between the blocks, using cells that have data in all conditions (Group 3 cells), same format. The blue wireframe shows the decision manifold for Slow blocks, and the red shows it for Fast blocks. To avoid clutter, we do not show the confidence intervals. (**D**) Same data as in (C) but projected into the space of PCs 2, 3, and 4. This facilitates comparison of the manifold and trajectories in the PC2-PC4 plane, as well as the contextual shift (blue double arrow) between the two blocks captured by PC3. Data and code available at https://doi.org/10.6084/m9.figshare.20805586. PC, principal component.

approximately constant in time, as shown in Fig 2). See S1 Movie for an animated version of this figure. As indicated in Fig 4A (dotted black arrows), in both block types, the neural activity evolves in a clockwise manner in the space of PC1 and PC4, passing over a region of deliberation until reaching a commitment state (purple ellipses), whereupon it rapidly moves to a movement-specific initiation subspace (green ellipses) and then turns back toward the starting point during movement execution. Some of these phenomena have previously been reported using neural space analyses of preparatory and movement-related activity in cortical regions during instructed reaching tasks with a single target [84–87,92–94]. Here, our task allows us to

examine in more detail what happens during the process of prolonged deliberation, while subjects are selecting among multiple targets.

In Fig 4A and 4B, we have drawn a gray wireframe around all of the points from the beginning of the trial until commitment time (280 ms before movement onset), across all trial types, thus defining the subspace within which deliberation occurs. This subspace resembles a triangular surface that is extended mostly in PC2 and PC4 and curved slightly into PC1. It is quite thin—for example, in the slow block, the value of Ψ (see Methods) is 0.235, which is roughly equivalent to a triangular sheet whose thickness is 1/45th of the length of each side. We call this the "decision manifold."

The flow of neural states upon the decision manifold is quite orderly, proceeding from bottom to top as time elapses and shifting left and right with the sensory evidence. For example, consider the misleading trials, in red, which clearly reveal the switch in sensory evidence. In effect, the flow of the neural state during deliberation resembles the temporal profile of evidence (Fig 1C) mapped onto that curved wireframe surface. The neural state continues to flow along the decision manifold until it reaches one of two edges (purple ellipses) at the time of commitment and then turns into PC1 and accelerates to rapidly flow along one of two paths, each corresponding to the choice taken, until movement initiation (green ellipses).

Fig 4C compares the decision manifolds between the two blocks, this time using only the cells that possess trials in all conditions (Group 3; see Methods). Note that in the space of PCs 1, 2, and 4 the shape of the manifold is quite similar, with just a slight upward shift of the Fast relative to the Slow block. Fig 4D plots the same data in the space of PCs 2, 3, and 4. This effectively flattens the trajectories shown in panel C into the PC2-PC4 plane, which captures how the neural state changes as a function of evidence and elapsing time within each block. The difference between the blocks is captured by the orthogonal shift along PC3. Similar shifts in neural space have been reported to capture learning in motor cortex [95].

## Region-specific dynamics

While the structure of the neural space computed across all neurons is interesting, it is still more informative to compare that structure across the different brain regions in which we recorded. Because the loading matrix produced by PCA provides coefficients that map each individual neuron's contribution to each PC, we can calculate the weighted average of each region separately, as shown in Fig 5.

Importantly, note that while PMd and M1 both strongly reflect the first four components described above, the other regions do not. In particular, dlPFC shows a clear relationship to evidence in PC2 and a slight block dependence in PC3, but it shows virtually no effect of elapsing time or commitment (both PC1 and PC4 are relatively flat). By contrast, in both GPe and GPi, PC1 shows a clear response at commitment, but PC2 shows no relationship to evidence until the decision is made. These findings suggest distinct functional roles for these regions.

We can use these same region-specific PCs to construct region-specific neural space trajectories. Fig 6A shows this for the dorsal premotor cortex, where we see a structure that is quite similar to what was shown in Fig 4A for all neurons (not surprisingly since the PMd population is the largest). As before, we see a triangular decision manifold that is relatively thin (Ψ = 0.343) and extends along PC2 and PC4. However, note that it initially strongly leans in the negative PC1 direction and then curves around just before commitment (see the side view shown in Fig 6A, right). Interestingly, when plotted in the space of PCs 1, 2, and 4, the PMd decision manifold is curved as if it lies on the surface of a sphere (Fig 6C, spherical fit $R^2$ = 0.65). In contrast, the decision manifold of M1 cortex (Fig 6B) is remarkably flat and thin (Ψ = 0.252) and leans into the positive PC1 direction. Nevertheless, the evolution of the neural state

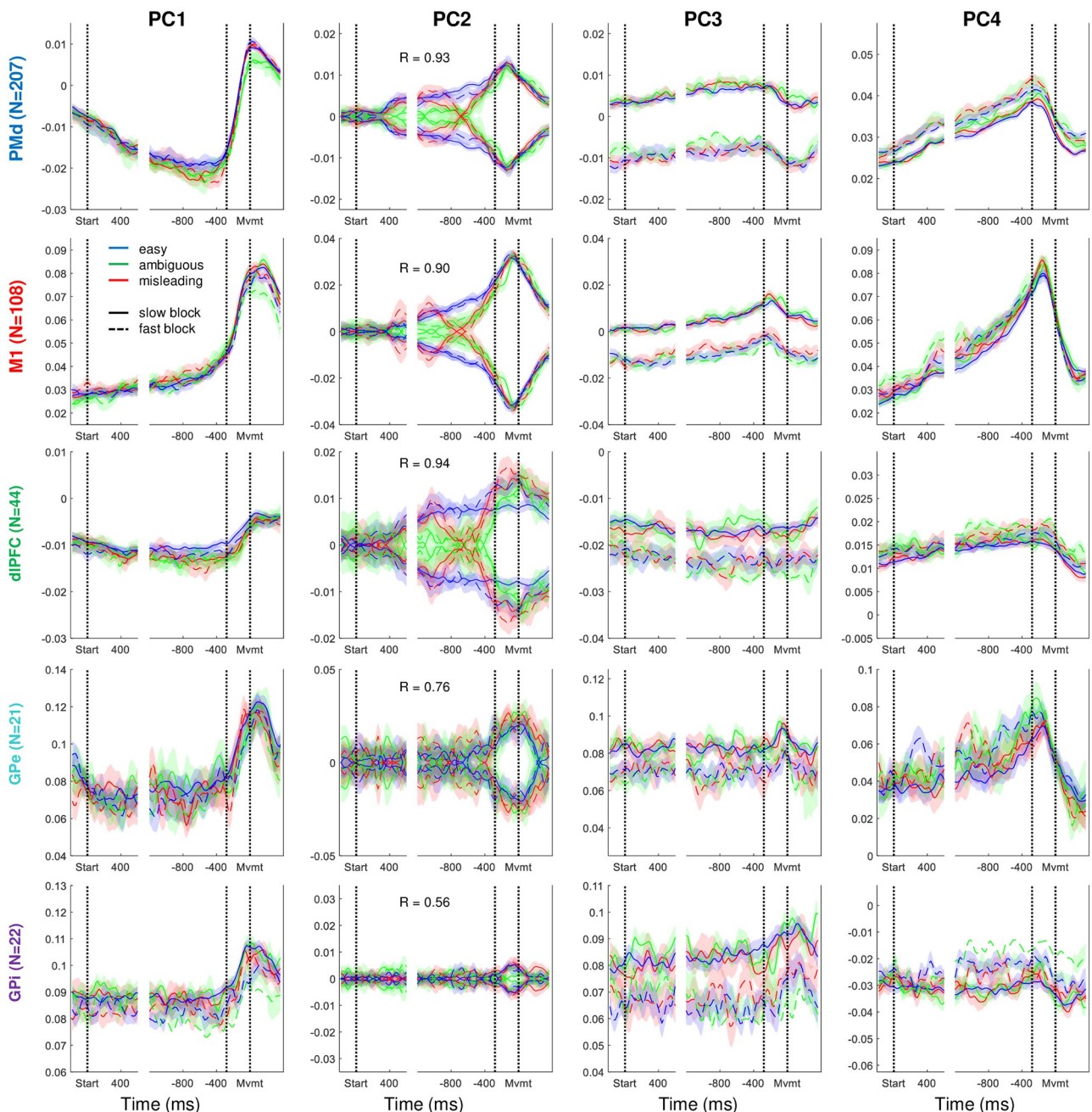

**Fig 5. The top four principal components shown in Fig 2, but here the weighted averages were computed separately from each of the five brain regions, using cells recorded in all conditions (Group 3).** For PC2, we report the R value of the correlation between evidence and neural activity during deliberation, as in Fig 3B. Data and code available at https://doi.org/10.6084/m9.figshare.20805586. dlPFC, dorsolateral prefrontal cortex; GPe, globus pallidus externus; GPi, globus pallidus internus; M1, primary motor cortex; PC, principal component; PMd, dorsal premotor cortex.

along the surface of the decision manifold in both regions obeys the same pattern shown in Fig 4, proceeding from bottom to top as time elapses and shifting left and right with the sensory evidence, always lying within the same subspace, curved for PMd and planar for M1.

As shown in Fig 6C, the curved PMd manifold fits well ($R^2 = 0.65$) to a small sphere whose radius is approximately half of the manifold range (that is, the manifold wraps around nearly

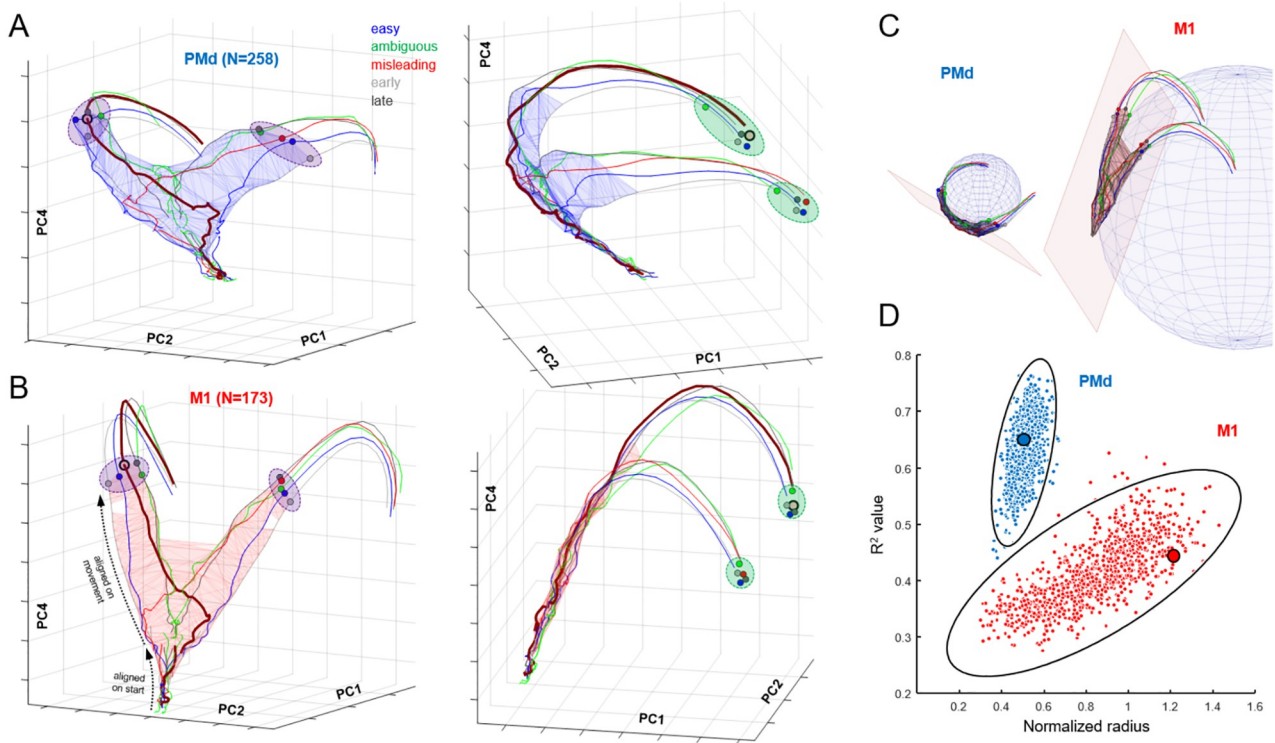

**Fig 6.** Neural trajectories in the space of PCs 1, 2, and 4, plotted using activity during Slow blocks (Group 1 cells), separately for PMd (**A**; see S2 Movie), and M1 (**B**; see S3 Movie). Separate trajectories are plotted for easy, ambiguous, and misleading trials, as well as all trials in which decisions were shorter than 1,400 ms (early) and in which decisions were longer than 1,400 ms (late). For clarity, confidence intervals have been omitted. As in Fig 4, trajectories are computed from data aligned on movement onset and extend from 1,400 ms prior. Blue (PMd) and red (M1) wireframes enclose all states before commitment (280 ms before movement). We also superimpose trajectories computed from data aligned on the start of the trial and until 500 ms later (projected through the same loading matrix). Dashed black arrows in panel B, left, indicate where these separate trajectories can be seen in the M1 space. In all panels, the trajectory of misleading trials in which the monkey correctly chose the right target is highlighted with a thicker line. Purple ellipses emphasize the time of commitment and green ellipses emphasize movement onset. (**C**) Spherical and planar fits to the decision manifolds computed from PMd (left) and M1 (right). (**D**) Comparison of spherical fits to bootstrapped data, assessed using the normalized radius and $R^2$ value of the best fit sphere. Solid lines indicate the 99% confidence ellipse for each resampled population, and large dots show the fits to the original data. Data and code available at https://doi.org/10.6084/m9.figshare.20805586. M1, primary motor cortex; PC, principal component; PMd, dorsal premotor cortex.

half of the sphere). By contrast, the planar M1 manifold produces a weaker fit ($R^2 = 0.44$) to a sphere with a much larger radius. To evaluate the robustness of this difference in manifold shapes, we performed a bootstrap analysis with 1,000 repetitions of spherical fits to the decision manifold (see Methods) using cells randomly resampled from the original PMd and M1 populations. As shown in Fig 6D, bootstrapped PMd manifolds consistently fit well to a small sphere. While bootstrapped M1 manifolds were sometimes best fitted with a small sphere, these fits were always weak ($R^2$ between 0.3 and 0.4). Importantly, no points from either distribution fell within the 99% confidence ellipse of the other distribution ($p < 0.001$), indicating that the difference in manifold shapes was highly robust. In the Discussion, we propose that these shapes reveal functionally meaningful differences between the neural dynamics of these two cortical regions.

It is also noteworthy that in both PMd and M1, the state reached at the moment of commitment (purple ellipses in Fig 6A and 6B) shows an orderly relationships with the reaction time in each trial type (shortest in "early" trials, and longest in "late" trials). This is in agreement with the observation that even at a single trial level, a consistent relationship between neural

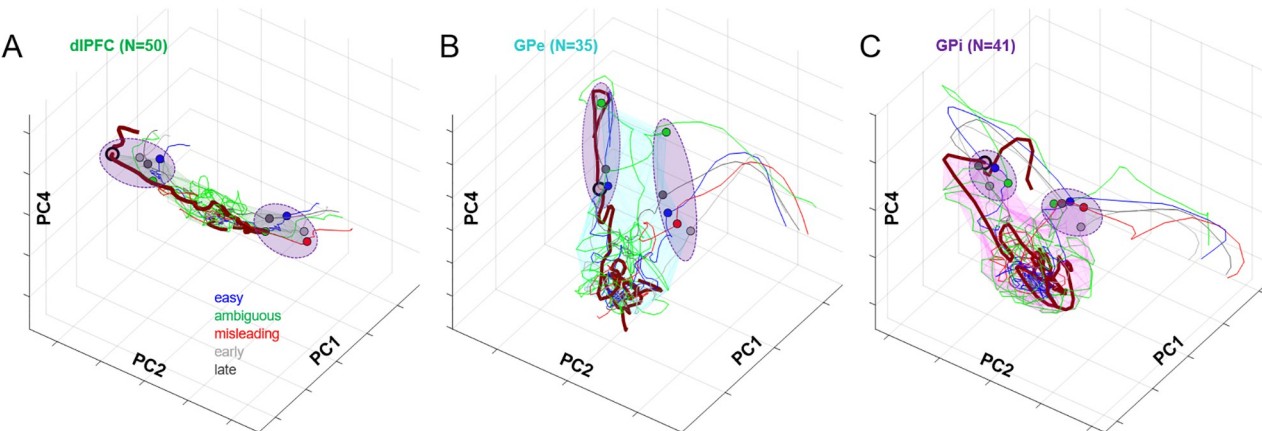

**Fig 7. Neural trajectories in the space of PCs 1, 2, and 4, plotted using activity during Slow blocks (Group 1 cells), separately for dlPFC (A), GPe (B), and GPi (C).** Same format as Fig 6. Data and code available at https://doi.org/10.6084/m9.figshare.20805586. dlPFC, dorsolateral prefrontal cortex; GPe, globus pallidus externus; GPi, globus pallidus internus; PC, principal component.

state and reaction time can be observed during a simple instructed reach task [84]. Furthermore, in M1, the trajectories after that point converge to arrive in a relatively compact subspace (green ellipses) at movement onset—what Churchland and colleagues [96] called an "optimal subspace."

In contrast to PMd and M1, the deliberation manifold in dlPFC (Fig 7A) is almost exclusively extended along PC2 ($\Psi = 0.564$, like a cylinder whose length is 22 times its radius). Like PMd and M1, the dlPFC neural state shifts left and right along PC2 with sensory evidence but shows almost no component of elapsing time (Fig 5), and after commitment, it exhibits only a small excursion into PC1. This suggests that neural activity in dlPFC primarily reflects the sensory evidence used to make decisions in the task, consistent with many previous studies [9,97–103].

A strong contrast to the cortical data is seen when we examine the neural state of cells recorded in the globus pallidus (GPe and GPi). In both regions, the neural state during deliberation is confined within a subspace that is not a thin manifold but instead resembles a ball compressed along PC2 and extended along PC4 (Fig 7B and 7C), which reflects the temporal component of urgency. During deliberation, activity in these regions tends to rise along PC4 but does not evolve in an orderly fashion along PC2, as seen in cortex. This does not appear to be caused by the lower number of cells recorded in GPe and GPi as compared to PMd and M1, because similar results hold when we restrict *all* regions to the same small number of cells (Fig Ca in S1 Text). Nevertheless, by the time commitment occurs, the state of both GPe and GPi lies in a choice-specific subspace (purple ellipses) and then evolves quickly to a corresponding initiation subspace. These findings are consistent with our previous report of GPe/GPi activity in the tokens task, in which we suggested that these regions do not determine the choice but rather contribute to the process of commitment [44].

## Analyses of the loading matrix

Another approach for inferring the putative functional contributions of different brain regions is to examine the distribution of the loading coefficients for cells in each region. For example, if a given population of cells is strongly related to the sensory evidence provided by token jumps, then cells in that population should tend to have higher magnitude loading onto PC2 than cells from another population that is less sensitive to evidence. As described in Methods,

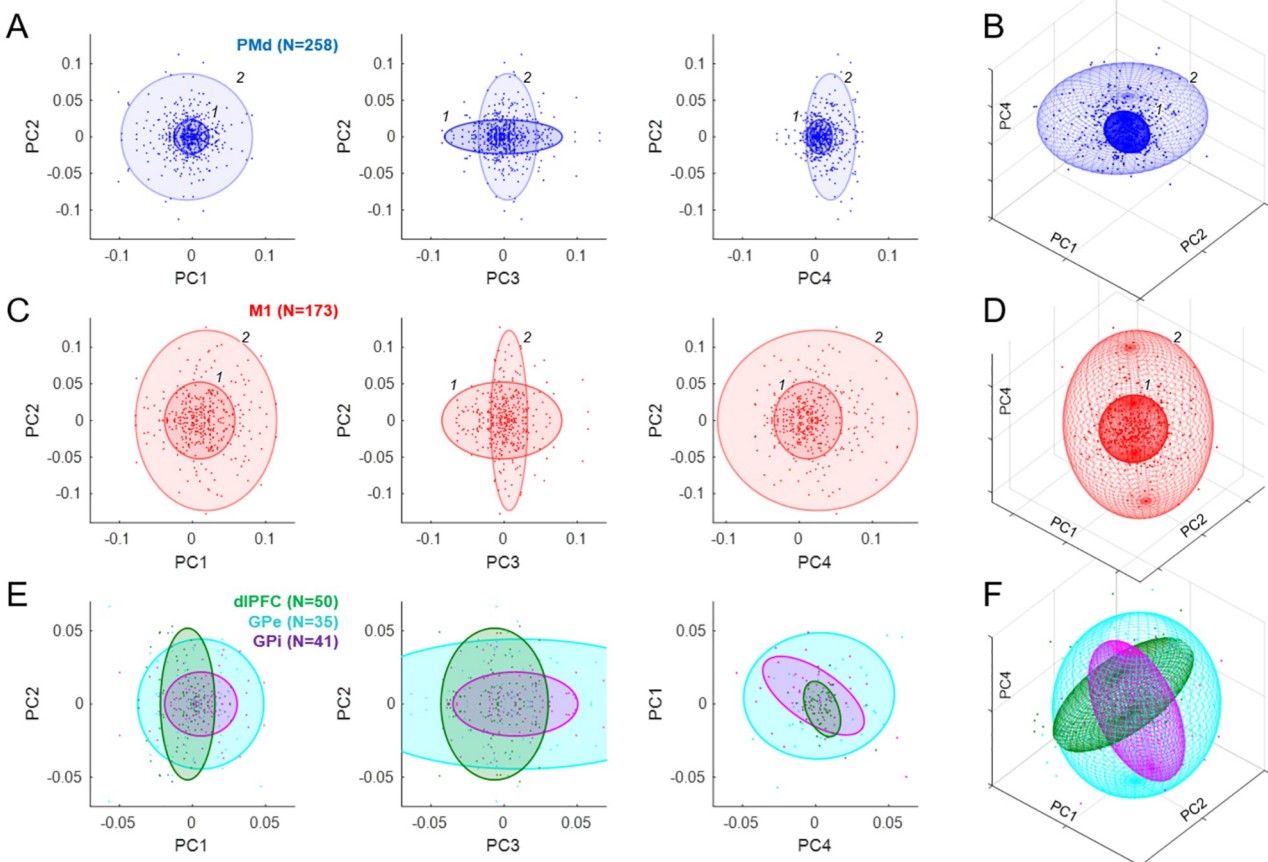

**Fig 8. Analysis of loading coefficients (all cells).** (**A**) Each point indicates the weight of the contribution of a given PMd cell to the principal component indicated on the axes. Colored ellipses indicate the centroid and 3 times standard deviation of each of the two gaussians (labeled *1* and *2*) that provided the best fit to the distribution of these populations. (**B**) The same PMd gaussians in the 3D space of PC1, PC2, and PC4. (**C**) Same as A for M1. (**D**) Same as B for M1. (**E**) Same as A for cell populations in dlPFC (green), GPe (cyan), and GPi (purple), each of which was best fit with a single gaussian. Note that unlike A and C, here the right panel shows PC4 vs. PC1. (**F**) Same as B for dlPFC, GPe, and GPi. Data and code available at https:// doi.org/10.6084/m9.figshare.20805586. dlPFC, dorsolateral prefrontal cortex; GPe, globus pallidus externus; GPi, globus pallidus internus; M1, primary motor cortex; PC, principal component; PMd, dorsal premotor cortex.

we characterized the distribution of loading coefficients for the first 11 PCs (which capture 95% of variance) by fitting the 11-dimensional space of points for each brain region with gaussian mixture models (GMMs) (for a related approach, see [77]). The results are shown in Fig 8.

The distribution of loading coefficients for cells in PMd (Fig 8A) was highly distributed but not without structure, and was best fit with two gaussians. The first (*1*: 68% contribution to the fit) was dominated by cells only weakly contributing to PCs 1, 2, and 4 but strongly to PC3. The second (*2*: 32%) included cells that strongly contributed to PCs 1 and 2 but more weakly to PCs 3 and 4. The distribution for M1 was also best fit with two gaussians, one (*1*: 56%) contributing mostly to PC3 and another (*2*: 44%) contributing to 1, 2, and 4 but not 3. Thus, in both PMd and M1, there was a trend for cells that most strongly reflect the animal's SAT (PC3) to be less strongly tuned to direction (PC2) and vice versa (Fig 8A and 8C, middle panels).

In contrast, populations in dlPFC, GPe, and GPi were well fit with a single gaussian (Fig 8E and 8F), though perhaps more structure would have been seen with larger numbers of cells.

Importantly, there were significant and potentially functionally relevant differences between these regions. In particular, the distribution of loadings from dlPFC and GPi were nearly orthogonal to each other in the space of PCs 1, 2, and 4 (Fig 8F). The dlPFC was extended along PC2 and not along the other components, consistent with the proposal that it primarily carries information on the sensory evidence provided by token movements. In contrast, GPi was relatively narrow in PC2 and instead extended along PCs 3 and 4, related to the block and time-dependent aspects of urgency. Furthermore, there was a significant negative correlation (R = −0.32, $p < 0.001$) between GPi loadings on PC1 versus PC4 (Fig 8E, right panel, purple). In other words, cells that build up over time (positive on PC4) tend to reduce their activity after commitment (negative on PC1), while cells that decrease over time (negative on PC4) tend to increase after commitment (positive on PC1).

In summary, our analyses of loading matrices suggest that while different regions do appear to contribute to different subsets of PCs (e.g., the orthogonal relationship of dlPFC and GPi in Fig 8F), the population within each region does not contain distinct clusters. In other words, while there is some structure in the loading matrix (e.g., Fig 8A and 8B, middle panel), the distributions of properties within each region are continuous.

Could that continuity of properties be an artifact resulting from dimensionality reduction? That is, if distinct categories of cells with different functional roles really did exist, would our analyses be able to identify them? To address this question, we created a variety of synthetic populations of neuron-like units and applied to them the same analyses we used to examine real data, including PCA and GMM analyses of the resulting loading matrix. As described in the S1 Text (see Fig F in S1 Text), this yielded three conclusions: It confirmed that PCA does correctly identify all of the components from which our synthetic neural populations were constructed. However, it also showed that some of the higher-order components (like PC5 in Fig 2) can result from PCA "cancelling out" some of the firing patterns already captured by lower-order components (e.g., PC2), in order to explain things like neurons tuned only during movement. Nevertheless, in all cases, the GMM analysis of the loading matrix correctly identified all of the actual categories of units in the synthetic populations. This suggests that the lack of distinct clusters we found in our neural data indeed reflects the absence of separate categories in the real neural populations. In S1 Text, we also discuss other methods that test for uniformity and clustering [75,76].

## Discussion

Many neurophysiological studies have suggested that decisions between actions unfold within the same sensorimotor regions responsible for the control of those actions [1,104–106]. This includes FEF and LIP for gaze choices [7,107–110] and PMd and MIP for reaching choices [8,36,110–114]. Several computational models of the decision mechanism suggest that it behaves like a "recurrent attractor system" [23–28], where reciprocally competing groups of neurons tuned for the available choices compete against each other until one group wins and the system falls into an attractor corresponding to a specific choice.

The results reported here provide strong support for this class of models. In particular, the cells we recorded in PMd and M1 do not appear to belong to separate categories related to decision-making versus movement preparation or execution but instead behave like part of an integrated dynamical system that implements a biased competition and transitions to commit to a choice through a winner-take-all process. During deliberation, the pattern of cell activity in these regions is confined to a highly constrained subspace in the shape of a thin manifold and is shifted around within that manifold by the decision variables pertinent to the task (here, the sensory evidence and the rising urgency). When commitment occurs, the same group of

cells now transitions from the decision manifold to a roughly orthogonal tube-shaped subspace corresponding to a specific choice [76] and then quickly flows to a subspace related to movement initiation [85,96]. This is precisely the kind of "winner-take-all" phase transition that occurs in recurrent attractor models. Importantly, the transition to commitment is not equivalent to the crossing of a threshold that is definable as the critical firing rate of a given neuron type, but instead resembles falling off the edge of the decision manifold into an orthogonal subspace that loses sensitivity to evidence. As can be seen in Fig 6A and 6B, that edge is oriented approximately diagonally with respect to PC2 and PC4, implying that straightforward summation of the activity of cells tuned to the chosen target may not be constant across trial types, as indeed often observed [36,115]. In other words, commitment is more nuanced than a threshold crossing; it is a state transition in a dynamical system [23].

Additional insights can be obtained by examining the low-dimensional components produced by dimensionality reduction. In particular, it is noteworthy that during deliberation, the four strongest components of neural activity (which together account for just over 80% of the variance) capture the key elements of the urgency-gating model [14]: the momentary evidence (in PC2), the urgency signal (a context-dependent baseline in PC3 and time-dependent ramping in PC4), and the transition to commitment (PC1). Perhaps even more important is how these components are differently expressed in the different cell populations. In particular, the neural state in dlPFC varies almost entirely along PC2 while, conversely, the state in GPe/GPi during deliberation is primarily determined by PC3 and PC4 and not at all by PC2. This suggests that information about sensory evidence is provided by prefrontal cortex [9,97–103] while the urgency signal is coming from the basal ganglia [44,116,117], and the two are combined in PMd and M1 to bias a competition between action choices [8,34,36,113,114,118].

The presence of the evidence-related component PC2 in the cortical data is particularly remarkable because the dimensionality reduction algorithm was not provided with any information about the variety of trial types (easy, ambiguous, etc.) that distinguish properties of deliberation but was merely given data averaged across four very large groups of trials: left or right choices during slow or fast blocks. Nevertheless, the difference of activity related to left versus right choices led the algorithm to assign a choice-related component that also happens to capture the evolving evidence for that choice. A neural population control method [91] verified that this is not a simple consequence of tuning. Additional analyses show that a unified evidence/choice component is obtained even if the PCA algorithm is only given data restricted to activity *after* movement onset (Fig Cb in S1 Text). This suggests consistency between cell properties before versus after movement onset [54], once again arguing against a categorical distinction between movement selection versus execution circuits and in favor of an integrated dynamical system, at least among the cells we recorded. Admittedly, the movements performed by our monkeys were quite simple and stereotyped, lacking the variation that would allow us to distinguish within our own data the neural components related to hand movement direction, joint angular velocities, muscle torques, or other execution-related variables. However, decades of previous neurophysiological recording studies have already established that neurons in PMd and M1 strongly reflect such variables [53,119–131] and are directly involved in the online guidance of reaching actions [132–135]. On the basis of such work, we therefore interpret the PMd/M1 activity we observed after movement onset as being related to the control of execution and not simply reflecting the target or goal.

The emergence of the other components is also highly robust. If the PCA algorithm is only given data from the slow block, PC3 is lost (unsurprisingly), but the other components remain (Fig Cd in S1 Text). Also not surprisingly, if we provide PCA with data from all 28 of our trial classes, the components are even more clearly distinguished, even though the number of cells that possess all of the required trials is reduced by 37% (Fig Ce in S1 Text). In fact, any

reasonable subset of data we have tried (e.g., from each animal separately) leads the PCA algorithm to identify the same functionally relevant components, albeit sometimes in a slightly different order depending on how much variance is captured by each. Finally, similar features are obtained if we leave out the cell duplication step (see Methods), although the resulting decision manifolds become asymmetrically distorted and some of the structure of loading matrices is more difficult to see (Fig E in S1 Text).

Finally, it is noteworthy that our findings reliably reproduce many previous observations from very different tasks, including ones in which monkeys did not have to make any decision at all and were simply instructed to reach out to a single target. This includes the general flow of the neural state from target presentation to movement onset and offset [85] (Fig 4), the orderly relationship between reaction times and the neural state in PMd/M1 [84] (Fig 6A and 6B, purple ellipses), the compact subspace in M1 at movement onset [96] (Fig 6A and 6B, green ellipses), and the presence of condition-independent components related to state transitions and elapsing time [87] (Fig 2, PC1, PC4). Although most of our data come from a prolonged period of deliberation that is not present in those other studies, the phenomena related to preparation and execution, which are shared between paradigms, are nevertheless robustly reproduced. This further strengthens the proposal that action selection and sensorimotor control are two modes of a single unified dynamical system.

Indeed, analyses of the loading matrix suggest that among the cells we recorded in PMd and M1, there is no categorical distinction between those involved in selection and those responsible for movement. Further analyses of synthetic populations (e.g., Fig F in S1 Text) demonstrate that some of the higher order PCs we observed (PC5 and PC6; see Fig 2) may result from the heterogeneity of properties across our cell population, which includes purely decision-related and purely movement-related cells. However, analysis of the loading matrix concluded that these properties are not clustered into distinct categories as in synthetic data (Fig Fd in S1 Text) but are instead distributed along a continuum (Fig 8).

While some of these findings could have been anticipated from analyses of individual cells, one important observation that would not have been possible concerns the shape of the decision subspace in the cortical populations. In particular, it is highly consistent across all of the different trial types and always resembles a thin manifold. This suggests strong normalization dynamics that are an inherent feature of recurrent attractor models. That is, the state of neural activity is pushed to lie on a surface that conserves some quantity (e.g., total neural activity with respect to some baseline) but is then free to move upon that surface under the influence of evidence, urgency, or simply noise. Furthermore, the particular shape of the manifold reveals consistent differences in the dynamics of different neural populations. In particular, regardless of what data we provide to PCA, the M1 manifold is always almost perfectly flat while the PMd manifold always exhibits a characteristic curvature. Interestingly, a very similar curvature was observed in preliminary analyses of PMd data in a very different decision task [136]. In contrast, the decision subspaces in the globus pallidus are much more compact and nothing like the thin manifolds in cortex.

Recent work suggests that the shape of the manifold on which neural states lie can reveal important features of the underlying dynamics of the system [70–72,137–142]. We believe the same holds for our data, particularly regarding the difference between the shapes of the decision manifolds in PMd versus M1. Notably, like many of our findings, this difference in manifold shapes is remarkably robust. It is observed whether or not we impose symmetry through cell duplication (Fig E in S1 Text, and S4 Movie), whether or not we use a square-root transform (Fig Cc in S1 Text), and whether we use data from each monkey separately (Fig D in S1 Text). It is significant at $p < 0.001$ according to a bootstrap test that randomly samples cells from each population (Fig 6D).

While the consistency of this difference in manifold shapes suggests that it reveals something important about the distinct neural dynamics of PMd versus M1, identifying the functional meaning of this difference is challenging. Many plausible possibilities exist, and here we take inspiration from previous work on computational models of recurrent neural circuits [23,26,28,143] to discuss interpretations that address two aspects of these shapes: curvature in the PC1-PC2 plane and curvature in the PC1-PC4 plane (Fig 6A and 6B).

We suggest that curvature in the PC1-PC2 plane reveals nonlinear interactions between competing cells in a given cortical region. As described in S1 Text, in a simple 2-neuron attractor model, the flow field of activity during deliberation is strongly constrained to lie within a narrow manifold whose shape is influenced by the function that governs mutual inhibition between competing cells (Fig G in S1 Text). If that function is flat for low inputs and then steeply rises for high inputs (approximating winner-take-all dynamics), then the manifold is flat as we observed in M1. In contrast, if the function resembles a shallow sigmoid, then the manifold is curved as we observed in PMd, and as others have reported in parietal cortex [137]. This could be related to the proposal, made in earlier modeling work [25], that PMd implements a gradual competition between groups of cells related to different movements, while M1 behaves more like a winner-take-all system.

In addition, we observed that the PMd manifold bends in the PC1-PC4 plane, shifting the direction of flow along the component associated with the transition from deciding to acting. Importantly, the timing of the shift foreshadows the moment of commitment, proceeding it by approximately 200 ms. We propose that these features could be the signature of a gradually emerging positive feedback in the cortico-striatal-thalamo-cortical circuit, which gradually overcomes inhibitory signals preventing premature selection [144]. This hypothesis would predict a correlation between how directional selectivity begins to emerge in GPi and how the PMd state begins to flow toward commitment. Indeed, as shown in Fig H in S1 Text, such a correlation is very strong in our data. In particular, the speed at which the PMd population moves toward the commitment subspace (derivative of PC1 computed from PMd) is significantly and positively correlated with the emerging directional selectivity in GPi (absolute value of PC2 computed from GPi), consistent with positive feedback between these regions [143,145,146]. Preliminary modeling work [145] suggests that such positive feedback emerges when a critical contrast is reached between opposing groups of cells in PMd. This predicts that microstimulation in cortical regions will disrupt that critical contrast and delay movement onset if delivered just before (but not after) the moment of commitment, as recently confirmed [147]. However, establishing the causal relationships in the full recurrent circuit will require many additional future studies, including simultaneous microstimulation in one region and recording in the other.

In conclusion, our analyses support the hypothesis that decisions between actions emerge as a competition within the sensorimotor system [8–10,25,107,109,110], which is governed by recurrent attractor dynamics [23–28], as supported by optogenetic stimulation studies in mice [148,149]. According to this model [145], the competition for reach selection occurs between cortical cell groups associated with different candidate reaching actions within arm-related regions of PMd and M1 and is biased by decision-related information coming at least in part from the prefrontal cortex [9]. As time passes, that competition is invigorated by a context-dependent urgency signal from the basal ganglia [44,116,117], which gradually amplifies the competitive dynamics in PMd/M1. As the contrast develops between the activity of cells voting for the different actions, a congruent bias is gradually induced in the striatum and pallidum, leading to a positive feedback that results in a winner-take-all process [44,143,146], which constitutes the commitment to an action choice. That process then brings the cortical system to a state suitable for initiating the selected action [96,150], setting into motion the "first cog" of a dynamical machine that controls our actions in the world [29–31].

## Methods

### Subjects and apparatus

Neural recordings were performed in two male rhesus monkeys (*Macaca mulatta; monkey S*: 4 to 9 years old, 5 to 9 kg; *monkey Z*: 4 to 6 years old, 4 to 7kg). Animals were implanted, under anesthesia and aseptic conditions, with a titanium head fixation post and recording chambers. The University of Montreal animal ethics committee approved surgery, testing procedure, and animal care.

Monkeys sat head-fixed in a custom primate chair and performed two planar reaching tasks using a vertically oriented cordless stylus whose position was recorded by a digitizing tablet (*CalComp*, 125 Hz). Their nonacting hand was restrained on an arm rest with Velcro bands. In most sessions, unconstrained eye movements were recorded using an infrared camera (*ASL*, 120 Hz). Stimuli and continuous cursor feedback were projected onto a mirror suspended between the monkey's gaze and the tablet, creating the illusion that they are in the plane of the tablet. Neural activity was recorded from the hemisphere contralateral to the acting hand with tungsten microelectrodes (1.0 to 1.5 MΩ, *Frederic Haer* and *Alpha-Omega Eng.*) moved with a computer-controlled microdrive (*NAN Instruments*) and electrophysiological signals were acquired with the AlphaLab system (*Alpha-Omega Eng.*). Spikes were sorted offline using Plexon Offline Sorter (*Plexon*). Electrodes were targeted based on 3D reconstructions (Brainsight, Rogue Research) using structural MRI images (Siemens 3.0 T). Behavioral data were collected with the software used to display the task (LabView, *National Instruments*).

### Behavioral task and classification of trial types

Monkeys were trained to perform the "tokens" task (Fig 1A) in which they are presented with one central starting circle (1.75 cm radius) and two peripheral target circles (1.75 cm radius, arranged at 180˚ around a 5-cm radius circle). The monkey begins each trial by placing a handle in the central circle, in which 15 small tokens are randomly arranged. The tokens then begin to jump, one-by-one every 200 ms ("pre-decision interval"), from the center to one of the two peripheral targets always oriented at 180˚ to each other with respect to the center. The monkey's task is to move the handle to the target that he believes will ultimately receive the majority of tokens. The monkey is allowed to make the decision as soon as he feels sufficiently confident and has 500 ms to bring the cursor into a target after leaving the center. When the monkey reaches a target, the remaining tokens move more quickly to their final targets ("post-decision interval," which was either 150 ms in "Slow" blocks or 50 ms in "Fast" blocks (in a few sessions, the post-decision interval was reduced to 20 ms in fast blocks). Once all tokens have jumped, visual feedback is provided to the monkey (the chosen target turns green for correct choices or red for error trials) and a drop of water or fruit juice is delivered for choosing the correct target. A 1,500-ms intertrial interval precedes the following trial. We alternated between Slow and Fast blocks after about 75 to 125 trials, typically several times in each recording session.

The monkeys were also trained to perform a delayed reach (DR) task (usually 30 to 48 trials per recording session). In this task, the monkey again begins by placing the cursor in the central circle containing the 15 tokens. Next, one of six peripheral targets is presented (1.75 cm radius, spaced at 60˚ intervals around a 5-cm radius circle) and after a variable delay (500 ± 100 ms), the 15 tokens simultaneously jump into that target. This "GO signal" instructs the monkey to move the handle to the target to receive a drop of juice. This task is used to determine cell's task response and tuning as well as the animal's mean reaction time (RT), used as an estimate of the total delays attributable to sensory processing and response

initiation. This quantity was then used to estimate the time of commitment in the tokens task. That is, the "decision time" (DT) in the tokens task was quantified as movement onset minus the mean RT from the DR task.

The tokens task allows us to calculate, at each moment in time, the "success probability" (SP) associated with choosing each target. To characterize the success probability profile for each trial, we calculated this quantity (with respect to the target ultimately chosen by the monkey) for each token jump (Fig 1B). For example, with a total of 15 tokens, if at a particular moment in time the right target contains $N_R$ tokens, the left contains $N_L$ tokens, and $N_C$ tokens remain in the center, then the probability that the target on the right will ultimately be the correct one (that is, the success probability of guessing right) is:

$$p(R|N_R, N_L, N_C) = \frac{N_C!}{2^{N_C}} \sum_{k=0}^{\min(N_C, 7-N_L)} \frac{1}{k!(N_C - k)!} \tag{1}$$

Although each token jump in every trial was completely random, we could classify a posteriori some specific classes of trials embedded in the fully random sequence (e.g., Fig 1C). In previous studies, we defined "easy," "ambiguous," and "misleading" trials on the basis of their success probability profile defined with respect to the first token jump. In contrast, because here we were primarily interested in examining activity with respect to commitment, we defined these trial types according to the success probability with respect to commitment time, estimated to be 280 ms before movement onset. A trial was classified as "easy" if the SP was above 0.5 five tokens before commitment, above 0.55 three tokens before, and above 0.65 at the time of commitment. A trial was classified as "ambiguous" if SP was between 0.35 and 0.65 at five and three tokens before commitment as well as at the time of commitment. A trial was classified as "misleading" if SP was below 0.5 five tokens before commitment and then above 0.5 at the time of commitment. A trial was classified as "other" if it did not meet any of these criteria. Note that these four classes are nonoverlapping. In the "Slow" block, we defined "early" trials as those where DT < 1,400 and "late" as those where DT ≥ 1,400 ms. In the "Fast" block, "early" trials were defined as those where DT < 950 and "late" as those where DT ≥ 950 ms.

In all tokens task trials, the targets were presented in opposite directions from the center, in two of six possible locations around the circle. Their placement was chosen according to the tuning of recorded cells, and in cases where a cell was not tuned, the leftmost and rightmost targets were used. Thus, in all cases, there was a target to the left of the center and one to the right (sometimes at an oblique angle). Here, we grouped all of these into two groups: We defined the three targets between 90° and 270° as "left" targets, and the other three as "right" targets, and did not examine directional tuning in any more detail.

For the analyses in the present report, we defined 28 task conditions as follows. First, we separated trials into those recorded during the "Slow" block and those recorded during the "Fast" block, and in each block, we split trials into those in which a "left" target was chosen by the monkey and those in which a "right" target was chosen. This yields 4 main groups: Slow-Left, Slow-Right, Fast-Left, and Fast-Right. Next, we split each of these four main groups into Easy, Ambiguous, Misleading, and Other, yielding another 16 conditions (Slow-Easy-Left, Slow-Easy-Right, Slow-Ambiguous-Left, etc.). Finally, we again split our four main groups into Early and Late, yielding another 8 conditions (Slow-Early-Left, Slow-Early-Right, Slow-Late-Left, etc.).

## Neural recording

Detailed methods for neural recording in PMd and M1 are described in Thura and Cisek [36], and for recording in the globus pallidus in Thura and Cisek [44]. Recordings in dlPFC used the same methods as PMd/M1 and focused on the region just dorsal to the caudal end of the principal sulcus (Fig 1D). We used 2 to 4 independently moveable electrodes for cortical recordings, and one electrode at a time for recordings in the globus pallidus. Thus, most of the cells whose activity is analyzed and reported here were not recorded simultaneously.

In all sessions, we focused on cells showing any change of activity in the tokens task, and monkeys were usually performing the task while we were searching for cells. When one or more task related cells were isolated, we ran a block of 30 to 48 trials of the DR task to determine spatial tuning and select a preferred target (PT) for each cell (that is, the target associated with the highest firing rate during one or more task epochs). Next, we ran blocks of tokens task trials using the PT of an isolated cell and the 180° opposite target (OT). We sometimes simultaneously recorded several task-related cells showing different spatial preferences, and since we always selected a single pair of targets, the actual best direction for all of the recorded cells was not always among these two. We usually started recording cells in the slow block because monkeys were more conservative in this condition. It was thus easier to assess cell properties online and more convenient to search for cells because fewer rewards were spent. When possible, cells were tested with multiple repetitions of slow and fast blocks to control for potential confounds related to evolving signals, elapsing time, and the monkey's fatigue or satiation (see Thura and Cisek [43] for control analyses on this question).

## Data analysis

A neuron was included in the analysis if it was recorded in both Slow and Fast blocks, thus including trials in each of the four main conditions: Slow-Left, Slow-Right, Fast-Left, and Fast-Right. This constraint was satisfied by 637 neurons out of the total 736 recorded across all brain regions (291 in PMd, 195 in M1, 53 in dlPFC, 45 in GPe, and 53 in GPi). For each neuron and each trial, neural activity was aligned to movement onset and the firing frequency was computed using partial spike intervals in ninety 20-ms bins from 1,400 ms before to 400 ms after movement onset. The firing rate was then square root transformed and smoothed using a 25-ms gaussian kernel. Finally, we imposed symmetry on our neural population by duplicating each of our neurons with an identical "anti-neuron" that has the same activity in all trials, but with the trial labels switched between left and right choices. We describe our rationale for including this step below in the section "Rationale for duplicating neurons."

To perform PCA of neural activity, we followed the procedure outlined in Fig A in S1 Text (steps 1 to 4). First, for each cell, we grouped trials into our four main classes (Slow-Left, Slow-Right, Fast-Left, and Fast-Right). In each group, neural activity was averaged together regardless of the type of trial (easy, ambiguous, misleading, or other) and regardless of whether the choice was correct or not, or the reaction time was early or late. For each of these four classes, we constructed a $1{,}274 \times 60$ matrix where rows are individual neurons and columns are time bins (from 1,000 ms before to 200 ms after movements onset). We then concatenated these four matrices into a $1{,}274 \times 240$ matrix on which we performed standard PCA (using the *pca* function in Matlab 2019b), yielding a matrix of weights ("loading coefficients") between cells and PCs as well as the variance explained by each PC. For further analysis, we only kept the top 20 PCs, which explained 97.9% of the total variance, though most of the interpretation will focus on the first four (total 80.3% of variance).

In addition to standard PCA, we also tried other dimensionality reduction methods, including factor analysis and GPFA [67,69], but these yielded almost identical results. This is not

surprising. For example, the major advantage of GPFA is that it jointly provides filtering with dimensionality reduction, which is critical when analyzing data from individual trials in which many neurons were recorded simultaneously. Our neurons were not recorded simultaneously, so we had to combine trials into similarity classes under the assumption that specific trial types were associated with similar neural activities on different occasions. This meant that we averaged activity across multiple similar trials, yielding much of the noise reduction that GPFA would have otherwise provided. Given the similarity of our results with different dimensionality reduction techniques, we chose to use PCA because it is the simplest, most widely known and well understood, and ultimately easiest to interpret.

Using the 1,274 × 20 loading matrix, we computed the average neural activity profile along each PC for a given trial condition and a given population of neurons by multiplying each neuron's activity in that condition by its loading coefficient for that PC, summing all of these together, and dividing by the number of neurons in that population. We calculated confidence intervals around this average neural activity by randomly resampling trials, with replacement within a given trial condition, 1,000 times for each neuron and time bin. We performed this process for all neurons together as well as for each of our neural populations separately (PMd, M1, dlPFC, GPe, and GPi).

Because not every neuron possessed a trial in each condition, potentially making it inappropriate to compare "PC space" trajectories from different conditions, we defined the following three groups of neurons designed to facilitate the analyses of primary interest. Group 1 consisted of neurons that possessed all conditions in the Slow block (Easy, Ambiguous, Misleading, Other, Early, and Late), for both Left and Right choices, as well as Early and Late conditions in the Fast block. This group was the focus of most of our analyses (e.g., Figs 4A, 6 and 7) and included a total of 557 cells (258 in PMd, 173 in M1, 50 in dlPFC, 35 in GPe, and 41 in GPi). Group 2 consisted of neurons that possessed Easy, Ambiguous, Misleading, and Other conditions in the Fast block. This group included 452 cells (226 in PMd, 126 in M1, 46 in dlPFC, and 27 each in GPe and GPi). Finally, Group 3 consisted of neurons that possessed trials in all conditions, including 402 cells (207 in PMd, 108 in M1, 44 in dlPFC, 21 in GPe, and 22 in GPi). We used Group 1 neurons for analyses restricted to data in the Slow block (e.g., Fig 4A), Group 2 neurons for analyses restricted to the Fast block (Fig 4B), and Group 3 for analyses that directly compared the two blocks (e.g., Fig 4C and 4D). Although these definitions were used to make quantitative comparisons most accurate, they did not have a strong impact on the qualitative aspects of our data, which were similar even if we restricted all analyses to Group 3, which was the most restricted.

Once we reconstructed the temporal profile along each PC for given conditions, we plotted each of them across time (e.g., Fig 2) and with respect to one another (e.g., Fig 4). To analyze the subspace visited by a neural population during the deliberation process, we examined PCs 1, 2, and 4 (see Fig 2) and constructed a 3D concave hull enclosing all of the neural states until the moment of commitment, for all conditions within a given block (using the *alphaShape* function in Matlab 2019b). For some of our brain regions, this subspace resembles a thin 2D sheet, which we call the "decision manifold." To quantify its thickness, we compared its surface area / volume ratio with that of a perfect sphere with equal volume, using the equation $\Psi = \frac{\pi^{1/3}(6V)^{2/3}}{A}$, where A is the surface area and V is the volume [151]. The quantity $\Psi$ is called "sphericity" and is a dimensionless scalar that ranges from 0 for a 2D surface to 1 for a perfect sphere (the geometric shape with the smallest surface area / volume ratio). For the reader's intuition, we note that an equal-sided cube of any size has $\Psi = 0.806$, while a square sheet whose thickness is 1/50th of the length of each side has $\Psi = 0.171$.

To characterize the loading matrix, we considered each cell as a point in a high-dimensional space defined by its loading coefficients for the top 11 PCs (which accounted for 95.2% of the

total variance). We then fit the resulting distribution of points for cells in a given brain region with GMMs using the Expectation–Maximization algorithm (*fitgmdist* function in Matlab 2019b). For each fit, we tried GMMs consisting of anywhere from one to six 11-dimensional gaussians, performing 100 randomly initialized fits for each, and then used the Bayesian information criterion (BIC) to select the best fitting model. We found that with 100 randomly initialized fits, we reliably found the same best solution each time.

## Rationale for duplicating neurons

The rationale for duplicating the neurons stems from the fact that our population is inevitably a highly sparse undersampling of the millions of neurons in the regions where we recorded. Importantly, we can assume that this undersampling is not symmetric with respect to the proportions of neurons that contribute to movements to the left versus right. For example, we might have in our sample many high-firing cells preferring leftward movements, but fewer high-firing cells preferring rightward movements. Applying PCA to such asymmetrically undersampled data will force the algorithm to try to capture this variance by aligning PCs to that asymmetry. We believe this is not informative. We can assume that our sampling is unlikely to be symmetric with respect to leftward or rightward movements, so it will not reflect any real asymmetry that may or may not exist in the brain. It will only distort our data in ways that will make interpretations more difficult.

   For this reason, just before performing PCA, we imposed symmetry on our data by using the "anti-neuron" approach classically used to produce population histograms of neural activity. In short, we assumed that for every given neuron that we recorded, there exists in the brain another neuron that we did not record, which has the same properties (same firing rate profile, same sensitivity to relevant variables, etc.) but has the opposite relationship with respect to the direction of movement. To implement this assumption, for each of our neurons, we created a "sister" neuron with the same firing data except with the trial labels switched between left and right choices. We did this for all neurons regardless of whether they are tuned or not. All of the analyses shown in the main paper apply this anti-neuron duplication, but in S1 Text, we show analogous results obtained without that step. As can be seen (Fig E in S1 Text), the only difference is that the PCs produced without anti-neuron duplication are slightly rotated and skewed versions of the ones produced after anti-neuron duplication, and their properties are slightly mixed between adjacent PCs. However, the general conclusions remain unchanged.

## Control analyses

To determine whether the PCs we found in the population data are a trivial consequence of properties of single neurons (e.g., directional tuning), we applied a control based on the TME method of Elsayed and Cunningham [91], using code they provide at htpps://github.com/gamaleldin/TME. Briefly, this method constructs surrogate data sets that preserve the primary first- and second-order covariance in the data along the temporal, neural, and condition-dependent dimensions but are otherwise maximally random. For example, tuning is preserved but not correlated across time. Any metric applied to analyzing the real data can then be compared to that same metric applied to surrogate data sets. If the metric computed from the real data lies within the distribution of that metric computed from surrogate data, then whatever is measured by the metric is a simple consequence of first- or second-order covariance.

   Here, we were interested to know whether our finding of PCs that covary with evidence is a simple consequence of cell tuning. To that end, we computed the correlation between the evidence provided by token jumps (e.g., Fig 1C) and the temporal profiles of the components generated by PCA when applied to surrogate data. Our metric was equal to the maximum

absolute value of the correlation coefficient for any of the top 10 components. We calculated this metric for the components produced using the real data (in which the best correlation was with PC2) and compared it to the distribution of the metric for 100 surrogate data sets generated using the TME method. That is, for each of the 100 surrogate data sets, we performed PCA on the four main trial classes and then used these to calculate the temporal profile of the top 10 PCs for all trial types, correlated each of these with evidence, and used the highest correlation value. We consider the emergence of an evidence-related PC as a nontrivial consequence of tuning if the correlation metric of fewer than 5 of these 100 surrogate data sets is equal to or higher than the metric applied to the real data (that is, $p < 0.05$).

To determine whether the shapes of PMd and M1 decision manifolds were significantly different from each other, we performed a comparison using spherical fitting. As shown for the original data in Fig 6C, a curved decision manifold (like in PMd) is best fit by a sphere with a relatively small radius, while a flat manifold (like in M1) is best fit by a sphere with a large radius. To yield a comparable scale for cell groups with different firing rates, we normalized the radius by the maximum range of the decision manifold. Next, to test the robustness of this difference in shapes, we performed a bootstrap analysis with 1,000 iterations. For each iteration, we randomly resampled the population of cells in each region, with replacement, and performed PCA using movement-aligned data from the slow block (that is, only Slow-Left and Slow-Right trials). Because we only used the slow block, no SAT-dependent PC (like PC3 in Fig 2) would be found, so the first three would consistently resemble our PCs 1, 2, and 4. Thus, we calculated the weighted average for the first three PCs, computed the decision manifold, and performed planar and spherical fits to the result. The results were then quantified as a distribution of normalized radius and coefficient of determination ($R^2$) values from the best fitting sphere. A high $R^2$ with a low radius was taken to indicate that the manifold was curved. To assess significance, 99% confidence ellipses were drawn around each distribution.

## Supporting information

**S1 Text. Supporting information. Fig A**. Schematic diagram of the analysis approach. Step 1: Neural activity is recorded from individual cells while the monkey performs individual trials of different types (e.g., "easy trial to the right," "misleading trial to the left," etc.), and the temporal profile is aligned on movement onset time. Step 2: For each cell, activity is averaged from all trials during the slow block in which the target on the right was chosen, to produce an average "Slow-R all" temporal profile. This is repeated for Slow-L, Fast-R, and Fast-L trial groups. Step 3: The four trial groups are concatenated into a single row for each cell, and the rows of all cells grouped into a large matrix. Step 4: Principal components analysis is performed on that matrix to produce the "loading matrix" L. Step 5: To generate the trajectory for a specific trial type and a specific population, we first average the neural activity of a given cell over all trials of the given type (e.g., misleading trials in the slow block in which the right target was chosen). Step 6: To compute the profile for that trial type and for a given principal component, the average activity of each cell is weighted by the corresponding row (cell) and column (PC) of the loading matrix L and summed. Step 7: The profile of each PC is divided by the number of cells used. Step 8: The resulting PC profiles are plotted against each other to produce neural trajectories. Here, the trajectory for Slow-Misleading-R trials is highlighted and superimposed over the trajectories of other slow block trials, as in Fig 4a. Note that steps 5–8 can be computed using arbitrary subsets of data (different trial groupings, different subsets of cells), and arbitrary alignment (on movement onset or on the first token jump), because all they depend upon is the loading matrix computed by steps 1–4. **Fig B**. The top four principal components produced when the loading matrix was computed from each brain region separately, same format as Fig 2. On the right are

shown the neural trajectories computed in the space of the components that reflect commitment, evidence, and elapsing time. Note that in some cases, the axis is inverted to facilitate comparison of these neural spaces to those shown in Figs 6 and 7. Data and code available at https://doi.org/10.6084/m9.figshare.20805586. **Fig C**. The top four principal components produced when the loading matrix was computed using only 35 cells per region (A), only data from movement onset to 400 ms later (B), data without square root transformation (C), only data from the slow block (D), or all data from all 28 trial conditions separately (E). Same format as Fig B. Data and code available at https://doi.org/10.6084/m9.figshare.20805586. **Fig D**. Results computed separately for each of the two animals. Panels A and B show the principal components (same format as Fig 2). Panels C and E show the top four components calculated using each brain region (similar to Fig 5). Panels D and F show the decision manifolds for PMd and M1 (similar to Fig 6). Data and code available at https://doi.org/10.6084/m9.figshare.20805586. **Fig E**. Results computed without the cell duplication step. (A) The top four principal components (similar to Fig 2). (B) The neural trajectories and decision manifold for PMd (similar to Fig 6a). (C) Neural trajectories and decision manifold for M1 (similar to Fig 2B). (D-F) Neural trajectories for dlPFC, GPe, and GPi (similar to Fig 7). (G) Loading distributions for PMd and M1 with respect to PC1 and PC2 (similar to Fig 8A and 8C, left). (H) Loading distributions for dlPFC, GPe, and GPi with respect to PC1, PC2, and PC4 (similar to Fig 8F). Data and code available at https://doi.org/10.6084/m9.figshare.20805586. **Fig F**. Analyses of synthetic populations. (A) The top 3 synthetic PCs (SPCs) obtained from a population of 600 neurons constructed using combinations of the original PCs 1, 2, and 4 (see Fig 2). (B) The loading coefficients of all 600 neurons, fitted with GMMs (gray ellipsoids). Colors indicate the three groups of neurons built with different combinations of those original PCs (see legend). (C) The top 3 SPCs obtained from a population of 600 neurons constructed only using the original PC1 and PC2 but separated into three distinct categories: cells that are only active during deliberation, cells only active during movement, and cells active during both epochs. (D) The loading coefficients of all 600 neurons, fitted with GMMs (gray ellipsoids). Colors indicate the three categories of neurons. Data and code available at https://doi.org/10.6084/m9.figshare.20805586. **Fig G**. Simulation of a simple 2-neuron recurrent competitive attractor. (A) The structure of the model is shown at left, next to the governing equations and parameter settings. Two forms of the interaction function $f(x)$ are shown in the inset at right. (B) The flow field for a system with high gain (G = 1) and steep slope (S = 15) when the input evidence is balanced such that $E_1 = E_2 = 0.2$. Blue arrows depict the flow and shading indicates the speed (yellow = fastest, dark blue = slowest). The purple outline indicates a region around the stable equilibrium, in which the state will tend to remain even with substantial noise. Note that the dynamics flow quickly toward that region and then move more slowly within it. (C) The stable regions of the same system for several input patterns, ranging from strong evidence in favor of choice 2 (top left, blue) to strong evidence in favor of choice 1 (bottom right, red). Note that these align in what is approximately a straight line across the state space (compare to dashed black line). (D) Same as B but for a system in which the interaction function has low gain (G = 0.1) and a shallow slope (S = 5). (E) Stable regions of the system in D. Note that now, the stable regions for the same range of input patterns do not form a straight line but rather fall on a curve roughly equivalent to a circle (dashed black curve). (F) Same as C and E but for models in which the input itself is constrained to lie on a curve. **Fig H**. Comparison of selectivity in GPi with the slope of the PMd decision manifold. (A) Comparison within six conditions: all Slow block trials, Slow block trials where DT < 1,400, Slow block trials where DT > 1,400, all Fast block trials, Fast block trials where DT < 950, Fast block trials where DT > 950. Each panel plots the absolute value of PC2 computed from GPi (red, left y-axis, shaded region indicates 95% CI) along with the derivative of PC1 computed from PMd (blue, right y-axis). The black vertical line indicates

the estimated time of commitment. At the top left corner are Pearson's R and the *p*-value of the correlation between these two signals from 1,000 ms to 280 ms before movement. (B) The correlation of these signals, from 1,000 to 280 ms before movement onset, across all 6 conditions. Data and code available at https://doi.org/10.6084/m9.figshare.20805586.
(PDF)

**S1 Movie. Neural trajectories in the space of PCs 1, 2, and 4, constructed separately for easy, ambiguous, and misleading trials as well as the other (unclassified) trials, using all cells recorded in the Slow block (Group 1 cells).**
(MP4)

**S2 Movie. Neural trajectories in the space of PCs 1, 2, and 4, plotted using PMd activity during Slow blocks (Group 1 cells).**
(MP4)

**S3 Movie. Neural trajectories in the space of PCs 1, 2, and 4, plotted using M1 activity during Slow blocks (Group 1 cells).**
(MP4)

**S4 Movie. Neural trajectories in the space of PCs 1, 2, and 4, constructed separately for easy, ambiguous, and misleading trials as well as the other (unclassified) trials, using all cells recorded in the Slow block (Group 1 cells), without the cell duplication step.**
(MP4)

## Acknowledgments

The authors wish to thank Terrence Sanger for valuable observations about our data, and John Kalaska and Andrea Green for additional observations and helpful comments on the manuscript.

## Author Contributions

**Conceptualization:** David Thura, Paul Cisek.

**Data curation:** David Thura, Paul Cisek.

**Formal analysis:** David Thura, Jean-François Cabana, Albert Feghaly, Paul Cisek.

**Funding acquisition:** Paul Cisek.

**Investigation:** David Thura, Paul Cisek.

**Methodology:** David Thura, Jean-François Cabana, Albert Feghaly, Paul Cisek.

**Project administration:** David Thura, Paul Cisek.

**Resources:** Paul Cisek.

**Software:** Jean-François Cabana, Albert Feghaly, Paul Cisek.

**Supervision:** David Thura, Paul Cisek.

**Validation:** David Thura, Jean-François Cabana, Albert Feghaly, Paul Cisek.

**Visualization:** David Thura, Jean-François Cabana, Albert Feghaly, Paul Cisek.

**Writing – original draft:** David Thura, Paul Cisek.

**Writing – review & editing:** David Thura, Paul Cisek.

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
