## [Editor Report · Decision Letter 0]

18 May 2022

Dear Drs Thura and Cisek, 

Thank you for submitting your manuscript entitled "Unified neural dynamics of decisions and actions in the cerebral cortex and basal ganglia" for consideration as a Research Article by PLOS Biology.

Your manuscript has now been evaluated by myself and an academic editor with relevant expertise. I am writing to let you know that we would like to send your submission out for external peer review. However, before we can send your manuscript to reviewers, we need you to complete your submission by providing the metadata that is required for full assessment. (See instructions below my signature.)

I also wanted to take a moment to (re)introduce myself. While it's been a while, you may recall me from our prior interactions when I was Deputy Editor at Neuron under Katja Brose. I’ve been freelance writing and editing for the past several years, and I recently started as the PLOS Biology Neuroscience Senior Editor/Section Manager. I am glad that you thought of PLOS Biology for this submission and I look forward to continued interactions with you during the peer review process.

With best wishes,

Kris

Kris Dickson, Ph.D. (she/her)

Neurosciences Senior Editor/Section Manager

PLOS Biology

kdickson@plos.org

***Full Submission instructions:

Please login to Editorial Manager where you will find the paper in the 'Submissions Needing Revisions' folder on your homepage. Please click 'Revise Submission' from the Action Links and complete all additional questions in the submission questionnaire.

Once your full submission is complete, your paper will undergo a series of checks in preparation for peer review. Once your manuscript has passed the checks it will be sent out for review. To provide the metadata for your submission, please Login to Editorial Manager (https://www.editorialmanager.com/pbiology) within two working days, i.e. by May 20 2022 11:59PM.

If your manuscript has been previously reviewed at another journal, PLOS Biology is willing to work with those reviews in order to avoid re-starting the process. Submission of the previous reviews is entirely optional and our ability to use them effectively will depend on the willingness of the previous journal to confirm the content of the reports and share the reviewer identities. Please note that we reserve the right to invite additional reviewers if we consider that additional/independent reviewers are needed, although we aim to avoid this as far as possible. In our experience, working with previous reviews does save time. 

If you would like to send previous reviewer reports to us, please email me at kdickson@plos.org to let me know, including the name of the previous journal and the manuscript ID the study was given, as well as attaching a point-by-point response to reviewers that details how you have or plan to address the reviewers' concerns. 

Feel free to email us at plosbiology@plos.org if you have any queries relating to uploading your full submission.

---

## [Decision Letter · Decision Letter 1]

1 Jul 2022

Dear David and Paul,

Thank you for your patience while your manuscript "Unified neural dynamics of decisions and actions in the cerebral cortex and basal ganglia" was peer-reviewed at PLOS Biology. It has now been evaluated by the PLOS Biology editors, an Academic Editor with relevant expertise, and by 2 independent reviewers. 

I am delighted to tell you that, based on the reviewer feedback, we are likely to accept this manuscript for publication. At this stage we simply ask that you address a few textual editorial requests, as well as the data and other policy-related requests listed at the bottom of this email.

Editorial requests:

TITLE:

We'd like to suggest a title change that we feel will make this work somewhat more accessible to our broad audience. Something along the lines of:

1) Sensorimotor decisions and actions reflect unified neural dynamics

2) Movement decisions and actions are mediated by similar neural dynamics

3) Movement decisions and actions are mediated by similar neural dynamics in the cerebral cortex and basal ganglia

ABSTRACT:

Our non-native English speaking editors felt that replacing "implemented" with "coded" or "regulated" or something similar would be preferable in the abstract.

BLURB for ETOC USE:

Please also provide a blurb which will be included in our weekly and monthly Electronic Table of Contents (eTOCs), sent out to readers of PLOS Biology. This blurb may also be used to promote your article on social media. The blurb should be about 30-40 words long and is subject to editorial changes. It should, without exaggeration, entice people to read your manuscript, should not be redundant with the title and should not contain acronyms or abbreviations. For examples, view our author guidelines: https://journals.plos.org/plosbiology/s/revising-your-manuscript#loc-blurb

We expect to receive your revised manuscript within two weeks. 

*Published Peer Review History*

*Press*

Sincerely,

Kris

Kris Dickson, Ph.D. (she/her)

Neurosciences Senior Editor/Section Manager,

kdickson@plos.org,

PLOS Biology

DATA POLICY:

Note that we do not require all raw data. Rather, we ask that all individual quantitative observations that underlie the data summarized in the figures and results of your paper be made available. While I note that you'd indicated that the data are included as supplemental files, I only saw supplemental movie files. 

Please add files, in one of the following forms, for the underlying data in:

Fig 2, Fig 3A-C, Fig 4A-D, Fig 5, Fig 6A,B,D, Fig 7A-C, Fig 8A-F

Supplemental Fig 2, Fig 3A-E, Fig 4A-F, Fig 5A-H, Fig 6A-D, Fig 7B-F, Fig 8A-B

1) As Supplementary files (e.g., excel). Please ensure that all data files are uploaded as 'Supporting Information' and are invariably referred to (in the manuscript, figure legends, and the Description field when uploading your files) using the following format verbatim: S1 Data, S2 Data, etc. Multiple panels of a single or even several figures can be included as multiple sheets in one excel file that is saved using exactly the following convention: S1_Data.xlsx (using an underscore).

2) As a deposition in a publicly available repository. Please also provide the accession code or a reviewer link so that we may view your data before publication. 

Regardless of the method selected, please ensure that you provide the individual numerical values that underlie the summary data displayed in these figure panels as they are essential for readers to assess your analysis and to reproduce your findings.

Please also ENSURE THAT THE FIGURE LEGENDS in your manuscript include information on where the underlying data can be found, and ensure your supplemental data file/s has a legend. (NB - this step is often forgotten!!!)

Please ensure that your Data Statement in the submission system also accurately describes where your data can be found.

CODE REQUESTS:

If applicable, we ask that you share publicly any code that you created and that directly relates to the results described in your article, following best practices for code sharing, such as:

- Deposition of your code in public repositories

- Citation of publicly-available code that is assigned a persistent identifier in your reference list

- Use of open source definition licenses for your code

DATA NOT SHOWN?

As applicable, please note that per journal policy, we do not allow the mention of "data not shown", "personal communication", "manuscript in preparation" or other references to data that is not publicly available or contained within this manuscript. Please either remove mention of any such data or provide figures presenting the results and the data underlying these mentions.

Reviewer remarks:

Reviewer's Responses to Questions

PLOS authors have the option to publish the peer review history of their article (what does this mean?). If published, this will include your full peer review and any attached files.

Reviewer #1: No

Reviewer #2: No

Reviewer #1: This is a very important paper and is likely to be transformative on the field. I believe that it will become regarded as a classic, and will have a dramatic effect on the way data are analyzed. I will provide a review, below, but I hope that the editors and authors will see that my comments are meant to improve the paper, not to criticize it. Because this journal does not seem to have a blind comments to editor section in the peer review interface, I will state here that I am strongly supportive of publication and, because I think that these results could influence a broader audience, the journal should consider a News and View-like piece to accompany the article. 

In short, this paper looks at the neuronal basis of decisions - an obviously important problem. The authors record in macaques in 5 regions (PMd, M1, DLPFC, GPi, and GPe) during a choice task. Rather than the normal practice of laserlike focus on a single neuron, and what can be inferred from them, the authors take a population-centered approach. In other words, they use the population doctrine not the single neuron doctrine. Specifically, they use PCA (although they use other methods with similar results in the Supplement). They find that:

* The first PC represents the transition from deliberation to commitment

* The second PC represents the accumulated evidence

* The third PC reflects the urgency, as determined by block

* The fourth PC reflects urgency as well

This tool, then lets them do some interesting things. For example, they are able to detect a new difference between PMd/M1 and DLPFC, and between these cortical regions and the GP structures. These differences are one that would not be visible with conventional single unit methods.

The primary importance of the paper comes from its entirely new perspective, one that helps us move past old ways of looking at how decision-making is organized, and towards new ones. Thus, the results, while important in and of themselves, are also important because they illustrate how to use population tools - even relatively simple ones - to think new thoughts about the neural basis of choice. Indeed, the paper has existed in poster form for several years, and the posters themselves have been passed around, and have become well-known among many scholars as major important works. It is gratifying to finally see them turned into a manuscript, which is sure to have an even larger impact.

Besides the perspective, the core empirical results of the paper are also important. First, the provide further support for - and elaboration of - the idea that action selection and decision rely not just on the same circuits, but instead are really two ways of thinking about the same process. But the most important and exciting result is that commitment to a decision is not a thresholding process, but rather an attractor process. This is a major change, and should engender a major rethink in the literature, so much of which is about thresholds. Beyond this, there are several other important results - the Discussion does a good job of summarizing and contextualizing them. This is, clearly, a major paper, that reflects a good deal of thought over many years. 

MINOR COMMENTS

Given the current writing, it is hard to link the author's underlying hypotheses and the results one would expect. For example, in the first section 'Dimensionality reduction of neural activity into principal components', the authors spend a lot of time discussing how they did PCA and comparative analyses (GPFA). This is fine. But these methodological details seem to be present in place of (1) what the hypothesis being tested is, and (2) what the reader should expect to see in the PCA results given those and alternative hypotheses. 

In the same 'Dimesionality reduction…' section, the authors bring special focus to PC2. I agree that it is interesting that it shows a split between "easy", "hard" and "ambiguous". But, should we have expected this given the hypothesized overlap between 'choice' and 'execution'? The connections between hypotheses and results are remain somewhat opaque. 

It seems useful to use the TME method here. But, it sems like this is being used in place of a (mixed) selectivity analysis. Are we to interpret this control as saying the selectivity is random and mixed across the population? If so, it seems best to show this with something like a pairs or epairs test; note, I realize the authors used a clustering of loading coefficients later. 

I think part of the claim with TME being made here is that 'no neuron subtype is solely responsible' for these dynamics seen in the PCA. But, how much of these neurons exhibit a pure, linear mixed or nonlinear mixed selectivity? How much of the PCA results are in fact due to dynamic changes in the nonlinear mixtures of units for task-relevant variables (see 5 next, also).

If the underlying components of the decision model are a priori known, why is PCA used rather than a focused method such as regression and their underlying subspaces? This would be more theory/hypothesis-driven. If using a regression approach, then the authors could introduce movement execution and choice regressors (and their interaction). Doing this would allow testing the claim they setout to do, but right now, it's unclear what constitutes 'execution' in the PCs; I don't see movement variables, except for left or right. If the authors meant for choice to == integration of evidence, and left/right to == execution, please state this more clearly and why left/right aren't just action rules or space codes. 

Analysis of the loading matrix: I think this analysis has potential to answer some of the authors main questions. However, the section is unclear because the expected results given the hypotheses aren't stated up front. It would help the reader to know what they should be looking for to follow along

The authors state (lines 352-353): "this suggests that the lack of distinct clusters….reflect absence of categories in the neural populations". Is this statement meant to convey that the distribution of neuron encoding approaches what many called random mixed selectivity (Raposo et al., 2014; Hirokawa et al., 2019)? If so, I think it would be best to state that clearly and cite those works too.

abstract: what does "clusters within each region" mean?

I realize that many of the results come from as much as 7 years ago. As such it is not surprising that many of the references are limited to an earlier time. I do think the paper would be improved by adding many of the important papers that come from the period of more recency.

Saxena and Cunningham's paper on the population doctrine should be cited somewhere. Other relevant reviews include Urai and Churchland and Ebitz and Hayden. 

Reviewer #2: While what Thura et al. present in this manuscript is not earthshattering new, the paper provides a very nice holistic view of the neural dynamics underlying a complex task, whose components were largely studied independently before. The authors trained monkeys to perform a visual perceptual decision-making task, which required the animals to report a choice through performing a hand movement. Neurons were recorded separately from a number of different brain areas, including prefrontal cortex, premotor and primary motor cortex, and the basal ganglia. Since the neurons were not recorded simultaneously, rather than taking a single-trial decoding approach, the authors relied on averaged neural activity from different classes of trials, separated by the animal's choice (left vs. right), time pressure on the current trial (slow vs. fast), and history of the sensory evidence (easy vs. ambiguous vs. misleading). The analysis presents the trials as a continuous dynamic process, including deliberation, choice commitment, movement preparation, and movement execution, in which the different brain areas engage in to varying degrees. The paper is very well written, and I don't have any major comments to add. I only have a number of smaller comments to consider:

1) Fig. 3 legend, "A. Dotted lines show the sensory evidence […]": There don't appear to be any dotted lines in this panel.

2) Fig. 4A, B: Is there a way to convince oneself that the final parts of the shown trajectories are related to active movement execution (rather than potentially just passively returning to a "neutral" state for the next trial)?

3) Lines 377-379, "Importantly, the transition to commitment is not equivalent to the crossing of a neural "threshold", but instead resembles falling off the edge of the decision manifold into an orthogonal subspace that loses sensitivity to evidence.": Why is it not possible to think of reaching this edge as reaching a decision threshold from the point of view of terminating the deliberation phase (ignoring, for the moment, what happens dynamically once the edge is reached)?

4) Lines 379-382, "As can be seen […] implying that straightforward summation of the activity of cells tuned to the chosen target may not be constant across trial types, as indeed often observed […].": What about the possibility of a time-variant decision criterion (as proposed in many more recent decision models)? Are the results inconsistent with such an idea?

---

## [Decision Letter · Decision Letter 2]

29 Sep 2022

Dear Dr Thura,

Thank you for the submission of your revised Research Article "Integrated neural dynamics of sensorimotor decisions and actions" for publication in PLOS Biology. On behalf of my colleagues and the Academic Editor, 

Alex Gail, I am pleased to say that we can in principle accept your manuscript for publication, provided you address any remaining formatting and reporting issues. These will be detailed in an email you should receive within 2-3 business days from our colleagues in the journal operations team; no action is required from you until then. Please note that we will not be able to formally accept your manuscript and schedule it for publication until you have completed any requested changes.

Additionally, when submitting the final version of your submission, Alex has suggested that you might want to consider rephrasing part of your newly added text to improve the accessibility of the paper - specifically regarding the text block attempting to link your hypothesis with specific predictions that should be seen with the PCA-based data analytic approach. Alex noted that the newly added text seems to be mostly phrased from a single neuron perspective ("neurons [...] will be distinct [...]" and "neurons that mix all [...] factors"), which he felt didn't quite achieve this goal. He argued that the main idea of the dynamical systems approach is to emphasize system dynamics in a low-dimensional space. Alex therefore thought that translating the two alternative hypotheses that you want to compare against each other into two clearly distinguishable outcomes regarding low-dimensional state trajectories (or associated PC spectra) might be a more compelling approach and would help to make the results easier to interpret. We therefore ask that you consider this additional minor revision to the final version of the study.

PRESS

We frequently collaborate with press offices. If your institution or institutions have a press office, please notify them about your upcoming paper at this point, to enable them to help maximize its impact. If the press office is planning to promote your findings, we would be grateful if they could coordinate with biologypress@plos.org. If you have previously opted in to the early version process, we ask that you notify us immediately of any press plans so that we may opt out on your behalf.

Sincerely, 

Kris

Kris Dickson, Ph.D. (she/her)

Neurosciences Senior Editor/Section Manager

PLOS Biology

kdickson@plos.org